# Genome-wide identiffcation of the WRKY gene family in poplar and the positive role of *PsnWRKY95* in response to cadmium stress

**Yuzhao Ma**[1☉], **Yiqi Liu**[1☉], **Fenglin Jia**[2], **Wanying Zhu**[1], **Guoyue Wang**[1], **Xiaojin Yang**[1], **Qing Guo**[1,3]*, **Hongbo Zhang**[1]*

**1** Heilongjiang University of Science and Technology, Harbin, China, **2** Chengdu Institute of Biology, Chinese Academy of Sciences, Chengdu, China, **3** State Key Laboratory of Tree Genetics and Breeding, Northeast Forestry University, Harbin, China

☉ These authors contributed equally to this work.

* guoqing@usth.edu.cn (QG); zhanghongbo37@usth.edu.cn (HZ)

## Abstract

WRKY is a crucial transcription factor family in plants, participating in a variety of physiological processes and stress responses. In this study, we identified 102 WRKY genes from the poplar genome, randomly distributed on 18 chromosomes and one scaffold, and classified them into three subgroups based on phylogenetic analysis. Members of the same subgroup form similar structures due to their shared relatively conservative domains. All poplar WRKY proteins are hydrophilic, located in the cell nucleus, and form target relationships with a large number of miRNAs, with their promoter containing a large number of stress defense elements. The expansion of the poplar WRKY family mainly occurs through segmental duplication, and they also have abundant cross-species collinearity. Based on RNA-Seq, we identified 83 *WRKYs* significantly respond to cadmium (Cd) stress. Subsequently, we conducted a study on *WRKY95*, which was significantly up-regulated in the roots, stems, and leaves under Cd stress. Under cadmium toxicity, *the plant* height of *PsnWRKY95*-overexpressing plants increased by 16%−26% compared with the wild type (WT), the root length increased by 12%−27% compared with WT, the peroxidase (POD) activity was 28%−51% higher than that of WT, the chlorophyll content increased by 15%−29% compared with WT, the malondialdehyde (MDA) content decreased by 13%−32% compared with WT, and the electrical conductivity decreased by 9%−20% compared with WT, with the expression levels of *POD* and *HMA1* in the overexpressing plants also being higher than those in WT. Results from yeast experiments demonstrated that PsnWRKY95 can improve Cd tolerance by specifically binding to cadmium (Cd) resistance element G-box, activating the reactive oxygen clearance ability and downstream target gene. This study comprehensively analyzes the basic data of *WRKYs*, identifying their response to Cd stress and

**Data availability statement:** All relevant data are within the manuscript and its Supporting Information files.

**Funding:** Heilongjiang University of Science and Technology Introduction of High-level Talents Scientific Research Start-up Project (HKD202219).

**Competing interests:** The authors have declared that no competing interests exist.

**Abbreviation:** Cd, Cadmium; HMT, Heavy metal toxicity; SA, Salicylic Acid; JA, Jasmonic Acid; ABA, Abscisic Acid; PC, Phytochelatin; ML, Maximum Likelihood; MEGA, Molecular Evolutionary Genetics Analysis; GFP, Green Fluorescent Protein; ROS, Reactive Oxygen Species; POD, Peroxidase; MDA, Malondialdehyde.

specifically analyzes the stress-resistant function of *PsnWRKY95*, providing clues for understanding the molecular mechanism of WRKYs resistance to Cd.

## Introduction

In the past few decades, due to human activities such as mining, smelting, sewage irrigation, and farmland fertilization, the content of cadmium (Cd) in the soil has rapidly increased, making it one of the most dangerous environmental pollutants [1]. Currently, more than 7.0% of the soil in China exceeds the standard for cadmium content, making the remediation of cadmium-contaminated soil an urgent matter. Utilizing genetic engineering technology to select cadmium-resistant plant varieties for bioremediation is a promising environmentally friendly technique for cadmium soil pollution control [2]. Therefore, understanding the mechanism of plant tolerance to cadmium can lead to the development of new quantitative and qualitative strategies to enhance plant tolerance to cadmium, which is of great importance. Under cadmium toxicity, plants will suppress biomass accumulation, limit growth and photosynthesis, experience chlorosis, and disturbances in water balance and nutrient absorption. To cope with cadmium poisoning, plants trigger various physiological and biochemical pathway involving many genes, reprogramming their physiological, biochemical, and morphological features to adapt the current adverse environment [3].

Transcription factors (TFs) are the most abundant gene regulators in multicellular organism genomes, involved in coordinating or individually regulating the expression of various stress response genes, modifying plant functional protein to play a remarkable role, and enabling them to adapt to the soil of heavy metals. TFs selectively bind to the cis-acting elements of their target gene promoter region, promoting TF oligomerization with other regulatory protein through protein-protein or domain-domain interactions, affecting the expression of many genes, thereby regulating plant function [4]. Extensive research indicates that TF families such as WRKY, bHLH, GRAS, MYB, AP2/ERF, Dof, bZIP, and DREB play a key role in responses to heavy metal stress. In apple, MdMYB306 complexes with MdERF114, directly binds the promoter of autophagy gene MdATG16, and activates autophagy to clear Cd-induced damaged organelles [5]. Up-regulation of the OsGASR1 gene enhances aluminum tolerance in rice [6]. Compared with Cd-NAC soybean mutants, overexpression of Cd-NAC in the roots of transgenic soybeans can improve cadmium tolerance, characterized by longer roots and higher biomass [7]. OBP3 interacts with bHLH transcription factor ILR3, enhancing its DNA-binding and transcriptional activation in *Arabidopsis*, thus positively regulating iron deficiency response genes [8].

WRKY, one of the most important transcription factor families, has a highly conserved domain composed of about 60 amino acids. Its N-terminal contains a specific conserved domain of WRKY (WRKYGQK) and the C-terminal has a zinc finger base sequence C2H2 (Cx4–5Cx22–23HxH) or C2HC (Cx7Cx23HxC) [9]. The first WRKY transcription factor, SPF1, was obtained from sweet potato (*Ipomoea batatas*) in 1994. With the development of whole genome sequencing technology, WRKY

transcription factors have been gradually found in more and more species, with 74 in *Arabidopsis*, 160 in *Triticum aestivum*, 102 in *Oryza sativa*, and 140 in *Zea mays* [10,11]. Many studies have shown that WRKY transcription factors, which can regulate many developmental and physiological process such as seed dormancy, seed germination, development, root formation, plant growth, aging, and responses to various biotic and abiotic stress, are referred to as versatile factor. In *Oryza sativa*, after knocking out *OsWRKY29*, the seeds would become dormant, and OsWRKY29 can specifically bind to the promoters of abscisic acid response genes OsVP1 and OsABF1 to down-regulate their expression, thereby shortening the seed dormancy period [12]. *AtWRKY23* regulates the growth of *Arabidopsis* roots by adjusting the distribution of auxin [13]. In wheat, TaWRKY51 can bind to the promoter of the ethylene biosynthesis gene TaACS to inhibit its transcription, down-regulating the biosynthesis of ethylene and promoting the formation of lateral roots [14]. Overexpression of *GmWRKY58*, *GmWRKY76* and *OsWRKY72* in *Arabidopsis* would cause transgenic strains to flower early [15,16]. *CsWRKY46* in *Cucumis sativus* mediates the GABA pathway to regulate a series of cold stress response genes, enhancing the cold tolerance of transgenic plants [17]. The *IbWRKY47* gene has been proven to positively regulate stress-related genes and significantly enhance the salt tolerance of *Ipomoea batatas* [18]. Overexpression of *CaWRKY27* from pepper in tobacco positively regulates its resistance to *Ralstonia solanacearum* infection by regulating Salicylic Acid (SA), Jasmonic Acid (JA), and ethylene-mediated signaling pathways [19]. In *Malus pumila*, salt tolerance regulation is related to MiR156/SPL, which up-regulates the salt-tolerant gene *MdWRKY100* [20]. *Fagopyrum esculentum FtWRKY46* improves salt stress tolerance by improving the reactive oxygen clearance system [21]. In addition, there is extensive research on plant WRKY genes in heavy metal resistance. The *AlWRKY46* gene mutant shows increased secretion of malic acid and reduced aluminum accumulation in root tips, improving aluminum resistance [22]. Han proved in *Arabidopsis* that WRKY is involved in regulating cadmium stress, where WRKY12 directly targets GSH1, indirectly inhibits the expression of genes related to Phytochelatin (PC) synthesis, and negatively regulates cadmium accumulation and tolerance [23]. Under cadmium stress, *WRKY45* endows *Arabidopsis* with cadmium tolerance, and WRKY12 inhibits the expression of *Vicia faba* GSH1 [24]. *Oryza sativa OWRKY22* and core transcription factor ART1 jointly act on OSFRDL4 expression and citrate secretion, promoting *Oryza sativa* aluminum tolerance [25]. RNA-seq analysis confirmed that the transcription of *Vicia faba WRKY45*, *WRKY28*, *WRKY30*, *WRKY50*, and *WRKY71* was induced by cadmium stress. *Malus pumila* WRKY11 up-regulates the expression of *MdHMA5* gene in transgenic plants, enhancing the copper resistance [26]. Zhang et al. found that cadmium-induced AtWRKY13 activated the expression of BCD and PDR8 (as a cadmium extrusion pump), leading to increased $H_2S$ production and enhanced plant cadmium tolerance [27]. ZmWRKY64 enhances cadmium tolerance by up-regulating ZmSRG7 transcription, maintaining ROS homeostasis [28].

Heavy metal toxicity (HMT) severely threatens global agriculture and crop yields, impairing plant physiology, seed germination, inducing oxidative stress, and inhibiting photosynthesis [29]. Cadmium, with a 10–30 year half-life in plants, accumulates excessively in roots causing browning, reduced length and dry weight, blocked lateral roots, and functional impairment, leading to root death. It exacerbates growth inhibition and necrosis by disrupting carbon fixation, reducing chlorophyll, and photosynthesis, while damaging antioxidant systems to trigger massive ROS, directly harming organelles and causing cell death [30–32]. Soil cadmium contamination also degrades produce quality, endangers human health, and undermines agricultural sustainability [33]. Resistance to heavy metal adversity is crucial for plant survival, especially perennial plants. They require longer growth cycles to demonstrate their unique resilience mechanism developed through natural long-term selection. *Populus simonii × P. nigra* is one of such perennial plants, with a wide distribution, strong cold resistance, high adaptability, large wood accumulation, and high heavy metal tolerance, and can serve as a plant for the remediation of heavy metal contaminated soil in Northeast China. Currently, there is less research on the specific physiological roles of poplar in response to cadmium stress. This study focuses on identifying the 102 members of the poplar WRKY family, analyzing their physicochemical properties, phylogenetic development, and gene structure, counting their promoter cis-acting elements, analyzing their chromosome distribution, gene evolutionary relationship and GO functional annotation, and correlating with their target miRNAs. Based on RNA-Seq, we explored the response of *WRKYs*

to cadmium stress, and identified the *PsnWRKY95* that is highly expressed. We confirmed its subcellular location and obtained overexpression *PsnWRKY95* tobacco through agrobacterium-mediated transformation. We validated its function based on growth and physiological indicators in a cadmium toxicity. At the same time, we examined the interaction between the PsnWRKY95 and Cd resistant cis-acting element G-box using yeast one-hybrid technology. This research contributes to understanding the cadmium resistance mechanism of poplar and provide a new perspective for the development of stress-resistant poplar varieties.

## Materials and methods

### Identification of WRKY family genes and conserved structural domains

In this study, we downloaded the model plant *Populus trichocarpa* genome and annotation from Ensembl Plants database (https://plants.ensembl.org). We obtained the Hidden Markov Model (HMM) for WRKY (PF03106) from Pfam database (https://pfam.xfam.org/) (E-value $< 1 \times 10^{-5}$). Using the poplar genome, we employed HMMER local BLASTP to retrieve poplar WRKY sequences, and further validated the accuracy of these sequences with the Pfam (https://pfam.xfam.org/) and NCBI-CDD (https://www.ncbi.nlm.nih.gov/cdd). Multiple sequence alignment was performed using CLUSTALX2.0, and the results were visualized with Web Logo and DNAMAN (version 8.0) [34]. We explored the physicochemical properties of PtWRKYs, including protein length, isoelectric point (PI), molecular weight (MW), instability index (II), grand average of hydropathy (GRAVY) and aliphatic index (AI), using the ExPASy proteomics server (https://web.expasy.org/protparam/) [35]. The subcellular localization of PtWRKYs protein were predicted using Wolf PSORT [36].

### Phylogenetic and gene structure analysis

To investigate the evolutionary relationship among WRKY family protein, we downloaded the sequences of 71 AtWRKYs from the TAIR (https://www.arabidopsis.org). An unrooted phylogenetic tree was constructed using the maximum likelihood (ML) method in Molecular Evolutionary Genetics Analysis (MEGA 7.0) software, with 1000 Bootstrap replications and Poisson correction. The visualization and annotation of the phylogenetic tree were performed using the JTT + G + I model and ITOL (https://itol.embl.de/) software. The motif analysis of PtWRKYs protein was performed using the MEME (http://meme-suite.org/meme/tools/meme) with default parameters and a maximum motif number set to 10 [37]. We utilized TBtools to identify the positions of the UTR, CDS, and the WRKY domain.

### Chromosomal localization and gene duplication analysis of *PtWRKYs*

The localization of WRKY genes on the poplar chromosomes was determined using TBtools. We combined multiple collinearity scanning tools (MCScanX) and BLASTP analysis to assess the tandem duplication, segmental duplication, and interspecies gene collinearity relationships of *PtWRKYs*. Additionally, we calculated the Ka/Ks substitution rates of the homologous gene pairs of PtWRKYs using the "Simple Ka/Ks Calculator" function within TBtools [38]. This tool employs the Nei-Gojobori algorithm to estimate synonymous (Ks) and non-synonymous (Ka) substitution rates from aligned coding sequences of segmentally duplicated PtWRKY gene pairs. We also estimated their evolutionary time using the formula $T = Ks/2\lambda$, where $\lambda = 9.1 \times 10^9$ [39].

### Analysis of cis-acting elements, GO functional annotation and miRNA

The PlantCARE database (http://bioinformatics.psb.ugent.be/webtools/plantcare/html/) was used to analyze the cis-acting elements in the 2000 bp upstream promoter regions of PtWRKYs. GO annotations for the target genes were obtained from Blast2GO. We utilized agriGO (http://bioinfo.cau.edu.cn/agriGO/) with the TopGO "elim" algorithm to perform functional enrichment analysis of *PtWRKYs* in biological processes, molecular functions, and cellular component [40]. Plant miRNA regulate target genes by complementary binding. In this study, we used the psRNA target server (http://plantgrn.noble.org/

psRNATarget) to predict potential miRNA targeting PtWRKYs based on default parameters [41]. The targeting relationship between PtWRKYs and miRNAs were visualized using Cytoscape v3.3 (http://www.cytoscape.org/).

## Plant treatment and *PtWRKYs* expression analysis

*Populus simonii × P. nigra* were derived from the experimental forests of Northeast Forestry University (China). The experimental conditions were set with a day/night temperature of 26°C/22°C, and relative humidity of 60% to 70%, with a 16-hour light/8-hour dark cycle. After two months of hydroponic growth, the cuttings developed new root and leave, and seedlings showing consistent growth were selected for subsequent experiments. We collected root and leaf samples from 12 soil-cultured poplar seedlings subjected to water and 150 μM $CdCl_2$ stress for 0, 12, 24, and 36 hours, with three replicate, 24 samples preserved in liquid nitrogen for temporal and spatial expression analysis. Fifteen hydroponic poplar seedlings were treated with 150 μM $CdCl_2$ for 24 hours, with water treatment as a control, grouping every five seedlings as one replicate, and samples from root, stem, and leave were collected, resulting in 18 samples stored in liquid nitrogen for cadmium stress expression analysis. After quality assessment of RNA-seq raw data using FastQC, Trimmomatic was used to remove low-quality reads and adapter sequences; clean reads were aligned to the *Populus trichocarpa* reference genome using HISAT2; feature Counts was employed to count reads at the gene level, and then the DESeq2 package in R software was used for differential expression analysis. DEGs were screened with the criteria of $|FC| \geq 2$ and p-value $\leq 0.05$. Gene expression plots were generated using Excel. A Venn diagram of the DEGs for PtWRKYs was generated using VENNY 2.1.0 (https://bioinfogp.cnb.csic.es/tools/venny/index.html). Twelve highly significant differential PtWRKYs were selected for RT-qPCR validation of their RNA-Seq results, using Actin as the reference gene (S1 Table).

## Cloning and sequence analysis of the *PsnWRKY95* gene

Total RNA was extracted from poplar cultured seedlings, and cDNA was obtained using the PrimeScript™ RT Reagent Kit (Takara, Dalian). Primers for *PsnWRKY95* were designed based on the transcript sequence, with the primer sequences listed in S2 Table. *PsnWRKY95* fragment was obtained through PCR, sequencing was conducted by Sangon Biotech (http://www.sangon.com/). Sequence alignment was performed using BLASTn (http://blast.ncbi.nlm.nih.gov/Blast.cgi). The conserved domain of PsnWRKY95 was analyzed using the NCBI CD-Search program (https://www.ncbi.nlm.nih.gov/Structure/cdd/wrpsb.cgi). The STRING database (https://string-db.org) was used to identify the protein-protein interaction network and functional annotations for PsnWRKY95.

## Subcellular localization analysis of the PsnWRKY95 protein

To verify the subcellular localization of PsnWRKY95, restriction sites SalI and SpeI were introduced at the 5' and 3' ends of the coding region of PsnWRKY95. Then cloned PsnWRKY95 into the pBI121-GFP vector; the primer sequences are listed in S2 Table. The fusion plasmid was subsequently introduced into agrobacterium tumefaciens GV3101, and one-month-old tobacco were selected for transient transformation. The agrobacterium containing pBI121-PsnWRKY95-GFP and pBI121-GFP was injected into the lower epidermis of the tobacco leaves and incubated in the dark for 24 hours. Green fluorescent signals were observed under a Zeiss laser confocal microscope.

## Interaction analysis of PsnWRKY95 with cadmium resistance element

G-box (CATGTG) of the cadmium responsive element were cloned in three tandem repeats into the pAbAi vector to create bait reporter vectors AB-G-box. Competent cells for AB-G-box-Y1H were prepared. The restriction sites NdeI and BamHI were introduced at the 5' and 3' ends of the CDS sequence of PsnWRKY95 respectively, and inserted into pGADT7-Rec to form AD-PsnWRKY95 (S2 Table). These fusion vectors were transformed into yeast cells containing the bait vector, and the cells were cultured on SD/-Leu/AbA (500 ng/mL) for 3–5 days. Positive clones were serially diluted and plated on solid medium for further analysis.

### Generation of *PsnWRKY95*-overexpressing tobacco and cadmium resistance analysis

One-month-old tobacco (*Nicotiana tabacum* L. cv. Petit Havana SR-1) were selected for transformation using the leaf disc method. After infecting the WT leaf with agrobacterium containing pBI121-PsnWRKY95-GFP for 8 minutes, the discs were placed on differentiation medium and incubated in dark for 2–3 days. They were then transferred to a selection differentiation medium to identify adventitious buds. 2 cm length resistant buds were excised and placed on rooting selection medium, with a photoperiod of 16 hours light and 8 hours dark at a temperature of 24 ± 1°C. The differentiation medium consisted of MS, 0.5 mg/L NAA, 0.5 mg/L 6-BA, and 0.5 mg/L GA3, while the rooting medium was 1/2 MS, with a selection concentration of 50 mg/L Kan. DNA was extracted from the resistant plants for molecular detection, with WT DNA as a negative control and plasmid DNA as a positive control. The expression levels of *PsnWRKY95* in the transgenic lines were quantified using RT-qPCR, with three replicates.

1.5 months aged transgenic lines (T-1, T-4, and T-5) and WT were cultivated in soil and irrigated with a 150 µM $CdCl_2$ for seven days, with water treatment serving as a control. After Cd stress treatment, plant growth state observed, and growth metrics such as plant height and root length were recorded. Leaves from different plant lines were collected for physiological measurements, including POD (peroxidase), chlorophyll content, electrolyte leakage rate and MDA (malondialdehyde). For POD assay: Fresh leaves were homogenized in extraction buffer (50 mM potassium phosphate, pH 7.8, 1 mM EDTA; 1% PVP). The homogenate was centrifuged at 4500 rpm for 30 min at 4°C, and the supernatant was used. POD activity was quantified by monitoring absorbance changes of guaiacol $H_2O_2$ reaction products at 420 nm. Chlorophyll extraction: Fresh leaves were immersed in 5 mL 80% acetone in the dark for 24 h, then re-extracted 5 times with 80% acetone until bleached. Extracts were combined and made up to 25 mL. Absorbance at 470, 645, and 663 nm was measured against 80% acetone. Contents were calculated as: Chlorophyll a (mg/g) = (12.21 × A663 − 2.81 × A645) × V/(W × 1000), Chlorophyll b (mg/g) = (20.13 × A645 − 5.03 × A663) × V/(W × 1000), Carotenoids (mg/g) = (1000 × A470 − 3.27 × Ca − 104 × Cb)/229, Total chlorophyll = Ca + Cb (V: extract volume; W: fresh weight; Ca, Cb: chlorophyll a, b contents). MDA determination: Fresh roots, stems, leaves were homogenized in 10 mL 10% TCA, centrifuged at 4500 rpm for 20 min. 3 mL supernatant + 3 mL 0.5% TBA was heated at 100°C for 15 min. Absorbance at 450, 532, 600 nm was measured. MDA (µmol/g FW) = [6.45×(A532 − A600) − 0.56 × A450] × V/FW (V: extract volume (L); FW: fresh weight). Electrical conductivity: Leaf discs (1 cm diameter, 18 pieces) were added to 50 mL $ddH_2O$, vacuumed for 15 min (S1). After boiling for 20 min and cooling, S2 was measured. Relative conductivity = S1/S2. Simultaneously, the expression levels of Cd stress related genes such as POD and HMA1 were analyzed using RT-qPCR, with relevant primers listed in S1 Table.

## Results

### Identification of WRKY family genes in poplar

Using the Hidden Markov Model (HMM) of WRKY (PF03106) as a query, we searched the poplar genome and ultimately identified 102 WRKY protein through multi-validation with Pfam and NCBI-CDD. We extracted the repeat regions of PtWRKY protein and performed multiple sequence alignment and visualization using DNAMAN and Web Logo (Fig 1). The results show that the conserved domain of PtWRKYs contain approximately 60 amino acids, a C2H2 zinc finger structure, and a highly conserved WRKY-specific domain (WRKYGQK), except for PtWRKY72. Based on their distribution on the poplar chromosomes, the PtWRKYs were named PtWRKY1 to PtWRKY102. The protein lengths of PtWRKYs range from 136 to 733aa, with molecular weight between 15,544.9 and 78,906 kDa, and isoelectric points ranging from 4.8801 to 10.3962. All PtWRKYs are hydrophilic protein, with Potri.001G002400.1 being the most hydrophilic (−1.34) and Potri.014G009500.1 being the least hydrophilic (−0.363). The instability index varies significantly among PtWRKYs, with PtWRKY3, PtWRKY28, and PtWRKY98 exhibiting lower values, indicating poorer stability. The aliphatic index, which reflects the relative content of the protein' fat side chains determined by the content of Ala, Val, Ile, and Leu, ranges from 31.85 to 82 in PtWRKYs. Based on Wolf PSORT subcellular localization results, most PtWRKY protein are primarily located in the nucleus, with a few members also present in chloroplasts, cytoplasm, and mitochondria (S3 Table).

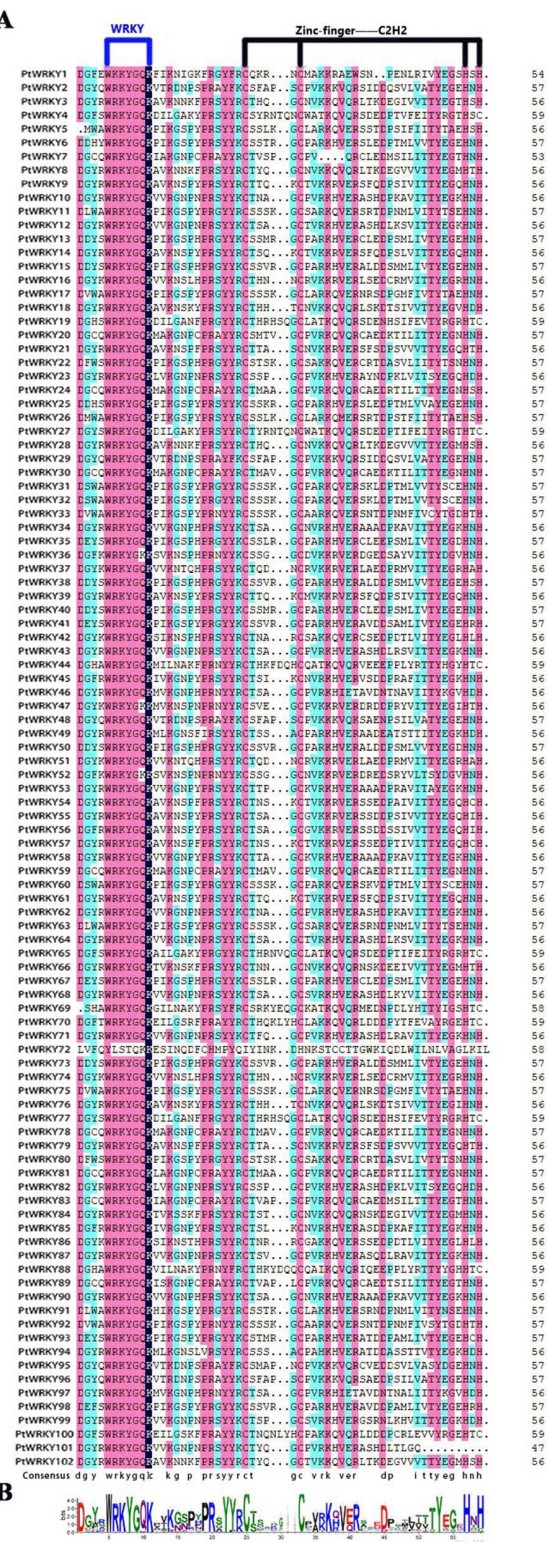

**Fig 1. Comparison of the WRKY domain in PtWRKYs.** (A) A multiple sequence alignment of PtWRKYs protein domains was performed using DNAMAN software, highlighting a conserved zinc finger structure (C2H2) and a WRKY domain (WRKYGQK). (B) Weblogo displays the sequence logo for the poplar WRKY domain (WRKYGQK). The total height of each column of letters indicates the degree of conservation at each position, while the height of each letter indicates the relative evaluation rate of the corresponding amino acid.

## Phylogenetic and sequence analysis

To explore the evolutionary relationship of WRKY, we constructed a phylogenetic tree using 102 PtWRKYs and 71 AtWRKYs (Fig 2 and S4 Table). Similar to *Arabidopsis thaliana*, the PtWRKYs were classified into Group I (22 members), Group II (55 members), and Group III (25 members), with Group II further divided into subgroup IIa, subgroup IIb, subgroup IIc, subgroup IId, and subgroup IIe, comprising 7, 9, 12, 13, and 14 members respectively. To compare the structural diversity of PtWRKYs, we analyzed their UTR, CDS, domains, and introns. Genes within the same group exhibit similar structures. Domain analysis indicates that all these genes possess a WRKY domain, with some containing additional specialized structures. Most genes in subgroup IId and PtWRKY60 contain the Plant_zn_clust domain, PtWRKY35 contains the TFCD_C superfamily domain, PtWRKY59 contains the Uso1_p115_C domain, PtWRKY70 contains the Nucleoporin_FG2 domain, PtWRKY78 contains the BRLZ domain, and PtWRKY98 contains the LGIC_ECD domain. There are obvious differences in intron patterns among PtWRKYs; Group I genes generally have 4–7 introns (PtWRKY85 having seven), Group II genes have 2–6 introns (PtWRKY83 and PtWRKY30 having six), and Group III genes possess 1–4 introns, with PtWRKY3, PtWRKY18, PtWRKY28, PtWRKY66, PtWRKY76, and PtWRKY84 having the least, only one each. We performed protein motif distribution analysis using MEME, revealing that each subgroup contains motif 1, motif 2, and motif 3. Group I have the most motifs, also including motif 4, motif 5, motif 6 and motif 10, while subgroup IIa, subgroup IIb, and subgroup IIc share motif 8, and subgroup IIc, subgroup IId, and subgroup IIe share motif 7; subgroup IIa, subgroup IIb, subgroup IId, and subgroup IIe share motif 9 (Fig 3). Interestingly, in addition to motif1, motif2, and motif3, Group III uniquely possesses motif5, suggesting specific function for these proteins.

## Analysis of chromosome localization and evolution with poplar WRKY genes

Based on the poplar genomic data, the 102 *PtWRKYs* are unevenly distributed across 18 chromosomes and one Scaffold_41. With the highest number (twelve) found on chromosome 1 and the fewest (only two) on chromosomes 12 and chromosomes 15. Interestingly, there are no WRKY gene on chromosome 9 (Fig 4A). The expansion of gene families in any species is primarily accomplished through segmental duplication events. Analysis through TBtools-McSxanX revealed that 101 *PtWRKYs* formed 160 segmental duplication pairs and one tandem duplication pairs, indicating that segmental duplication is a major factor in the expansion in poplar. To further investigate the evolutionary constraints of *PtWRKYs*, we analyzed the non-synonymous substitution rates (Ka), synonymous substitution rates (Ks), and Ka/Ks ratios of 160 pairs of homologous *PtWRKY* (S5 Table). The non-synonymous substitution rate Ka indicates amino acid changes due to base substitutions, with Ka values for the *PtWRKY* duplication pairs ranging from 0.0702 to 0.9344, suggesting that a large-scale gene duplication event occurred around 33 million years ago, with the most recent event occurring approximately 1.08 million years ago. Generally, Ka/Ks ratios less than, equal to, or greater than 1.0 indicate purifying selection, neutral selection, and positive selection, respectively. In this study, all 160 *PtWRKY* gene pairs had Ka/Ks values less than 1, indicating that the WRKY family in poplar has undergone strong purifying selection, thus reducing subsequent deleterious mutations.

To understand the syntenic relationship of WRKY family members among poplar, *Arabidopsis*, and *Oryza sativa*, we analyzed the collinearity between these species. 72 *PtWRKYs*, 38 *AtWRKYs* and 18 *OsWRKYs* constitute 104 collinear gene pairs, including 72 *PtWRKYs-AtWRKYs* and 32 *PtWRKYs-OsWRKYs* (Fig 4B and S6 Table). Each *PtWRKY* corresponds to one or more genes from other species; for instance, *PtWRKY4* corresponds to *AtWRKY46* (*AT2G46400.1*) and *OsWRKY15* (*Os01g0656400*); *PtWRKY31* corresponds to *AtWRKY65* (*AT1G29280.1*) and *OsWRKY13* (*Os01g0750100*); *PtWRKY65* corresponds to *AtWRKY30* (*AT5G24110.1*) and *OsWRKY15* (*Os01g0656400*); and *PtWRKY95* corresponds to *AtWRKY40* (*AT1G80840.1*) and *OsWRKY71* (*Os02g0181300*). This indicates that these homologous genes may have diverged from a common ancestor, retaining similar function.

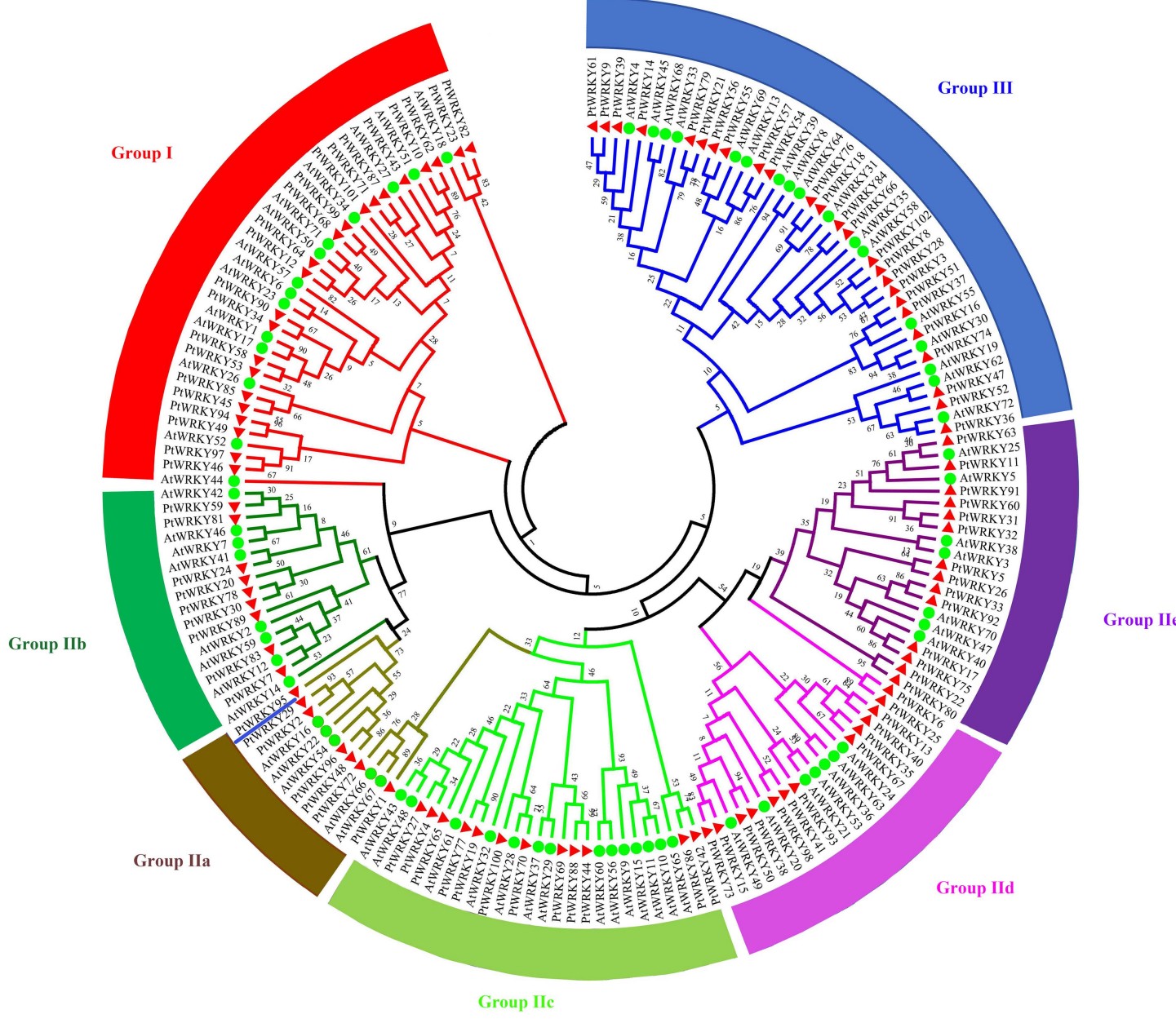

**Fig 2. Phylogenetic tree of PtWRKYs and AtWRKYs.** An unrooted phylogenetic tree of PtWRKYs and AtWRKYs. WRKY protein sequences was drawn using the Maximum Likelihood (ML) method in MEGA 7.0. The phylogenetic tree is divided into three subgroups: Group I, Group II, and Group III, where Group II is further divided into subgroup IIa, subgroup IIb, subgroup IIc, subgroup IId, and subgroup IIe. Each group is distinguished by a specific color, PtWRKYs are marked with red triangle, and AtWRKYs are marked with green circle.

## Analysis of cis-acting elements and GO functional annotation

Cis-acting elements play significant role in the transcriptional regulation of genes during plant development and in response to environmental stresses. In this study, we extracted the upstream 2000 bp promoter sequences of PtWRKYs to analyze their cis-acting elements using the PlantCARE. In addition to basic cis-acting elements such as the TATA-box

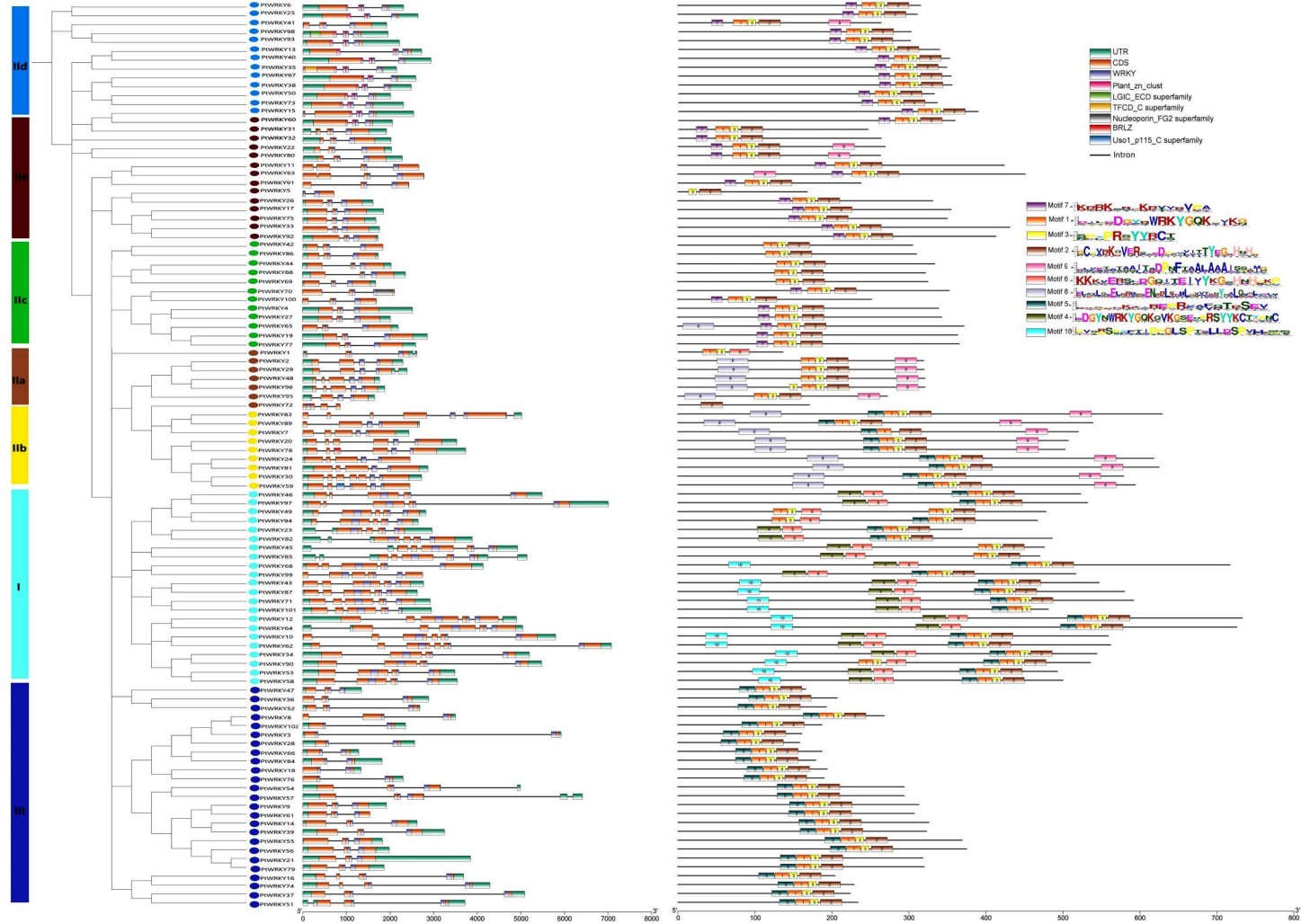

**Fig 3. PtWRKY gene structure and Motif.** A phylogenetic tree was constructed based on 102 PtWRKY proteins and divided into 3 subgroups. Dark green represents UTR, orange represents CDS, purple represents the WRKY domain, pink represents the Plant_zn_clust domain, light green represents the LGIC-ECD superfamily domain, yellow represents the TFCD_C superfamily domain, grey represents the Nucleoporin_FG2 domain, red represents the BRLZ domain, blue represents the Uso1_p115_C domain, and the "-" represents introns. The motifs were predicted using the MEME method, the numbers (1-10) in the colored rectangles represent motif 1-motif 10, and the length of the rectangle represents the size of the motif.

and CAAT-box, a lot of important cis-acting elements were identified, categorized by function into three groups: growth and development, hormone response, and abiotic stress response. The growth and development category includes the elements specific for seed and endosperm expression and the elements regulatory for palisade mesophyll cell differentiation. The hormone regulation category encompasses salicylic acid response elements, abscisic acid response elements, auxin response elements, and anthocyanin biosynthesis elements. The abiotic stress category includes stress response elements, defense response elements, low-temperature response elements, hypoxia-inducible elements, drought-inducible elements, and wound response elements (Fig 5 and S7 Table). These findings suggest that PtWRKYs possibly respond to environmental changes and regulate critical physiological and developmental processes through a series of stress elements and hormonal regulations. Functional analysis of the *PtWRKYs* by the GO online tool, the results indicate that the *PtWRKYs* are primarily involved in molecular functions, cellular components, and biological processes. These

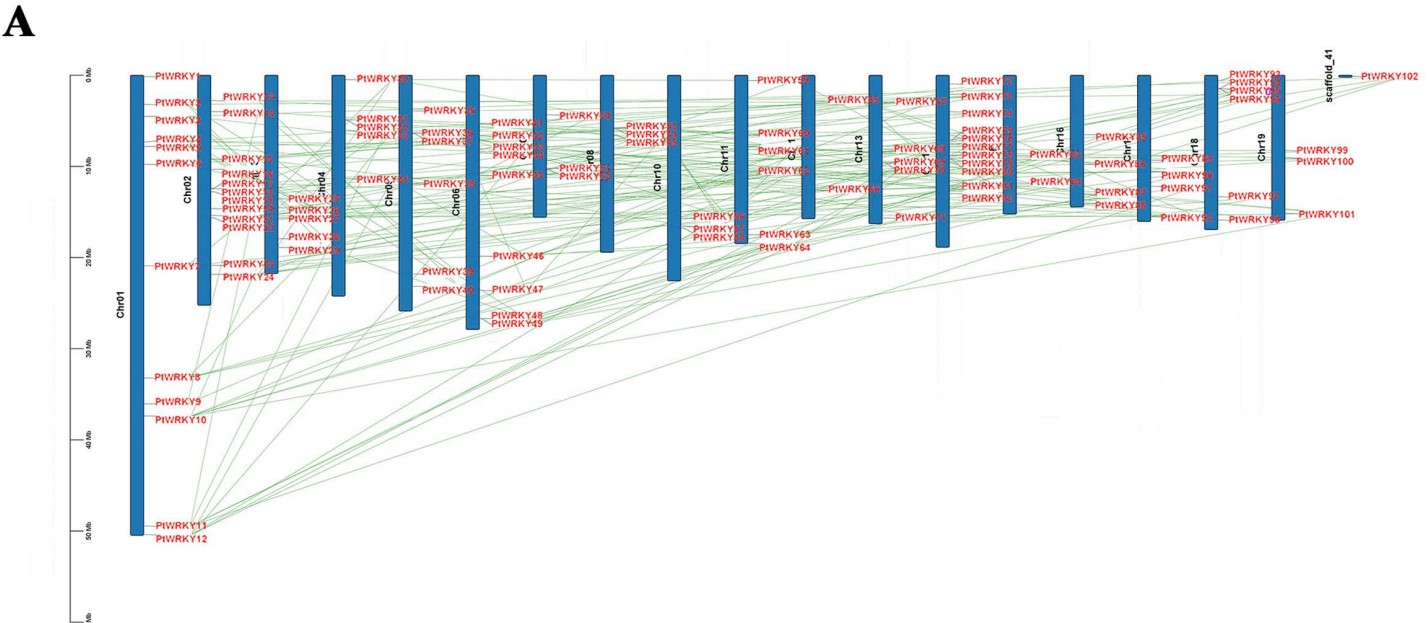

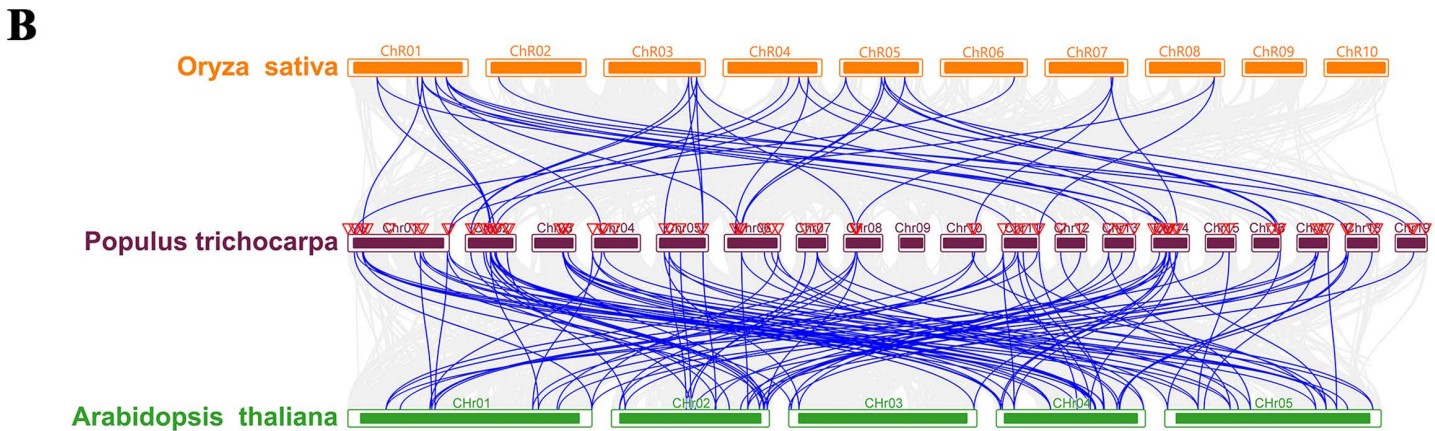

**Fig 4. Chromosomal localization and gene evolutionary relationship of *PtWRKYs*.** (A) The distribution of 102 WRKY genes on the poplar chromosomes, segmental duplications are represented by green lines, and tandem duplications are represented by purplish-red lines. (B) The collinearity relationship of *PtWRKYs*, *AtWRKYs*, and *OsWRKYs*, the red triangles represent poplar WRKY genes, and the blue lines represent orthologous gene pairs.

include response to abscisic acid (GO:0009737), salicylic acid (GO:0009751), jasmonic acid (GO:0009753), drought (GO:0009414), cold (GO:0009409), protein binding (GO:0005515), and transcription regulation region DNA binding (GO:0044212) (Fig 6).

## Target miRNAs of PtWRKYs

MicroRNA (miRNA) are a class of endogenous non-coding small RNA that regulate gene expression by specifically binding to target genes, participating in various physiological processes. The results of this study show that 91 PtWRKYs and

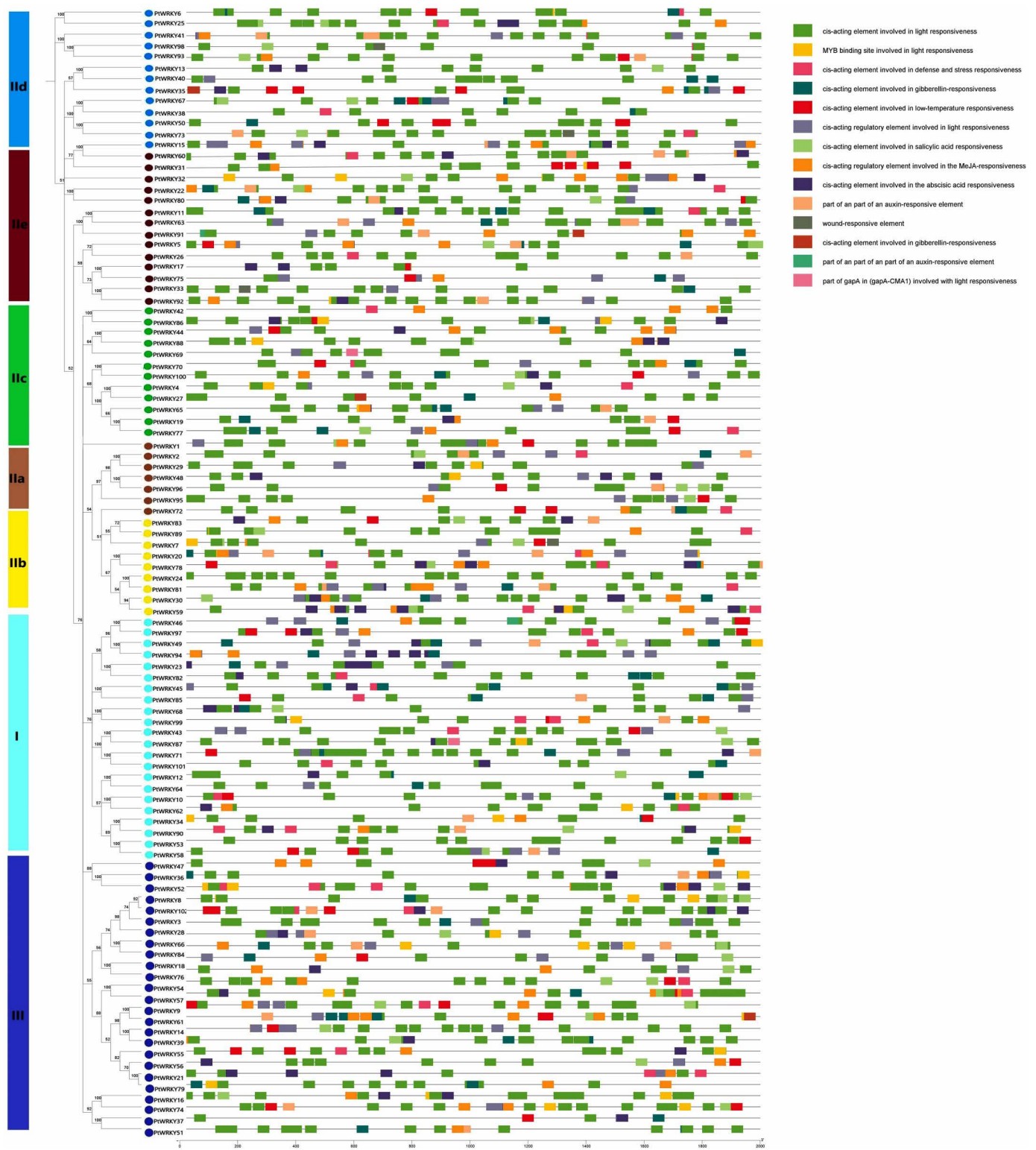

**Fig 5. Cis-acting element analysis of PtWRKYs promoter.** Analysis was performed on the upstream 2000 bp sequence of the 102 PtWRKYs, with different colored rectangles representing cis-acting elements with different function.

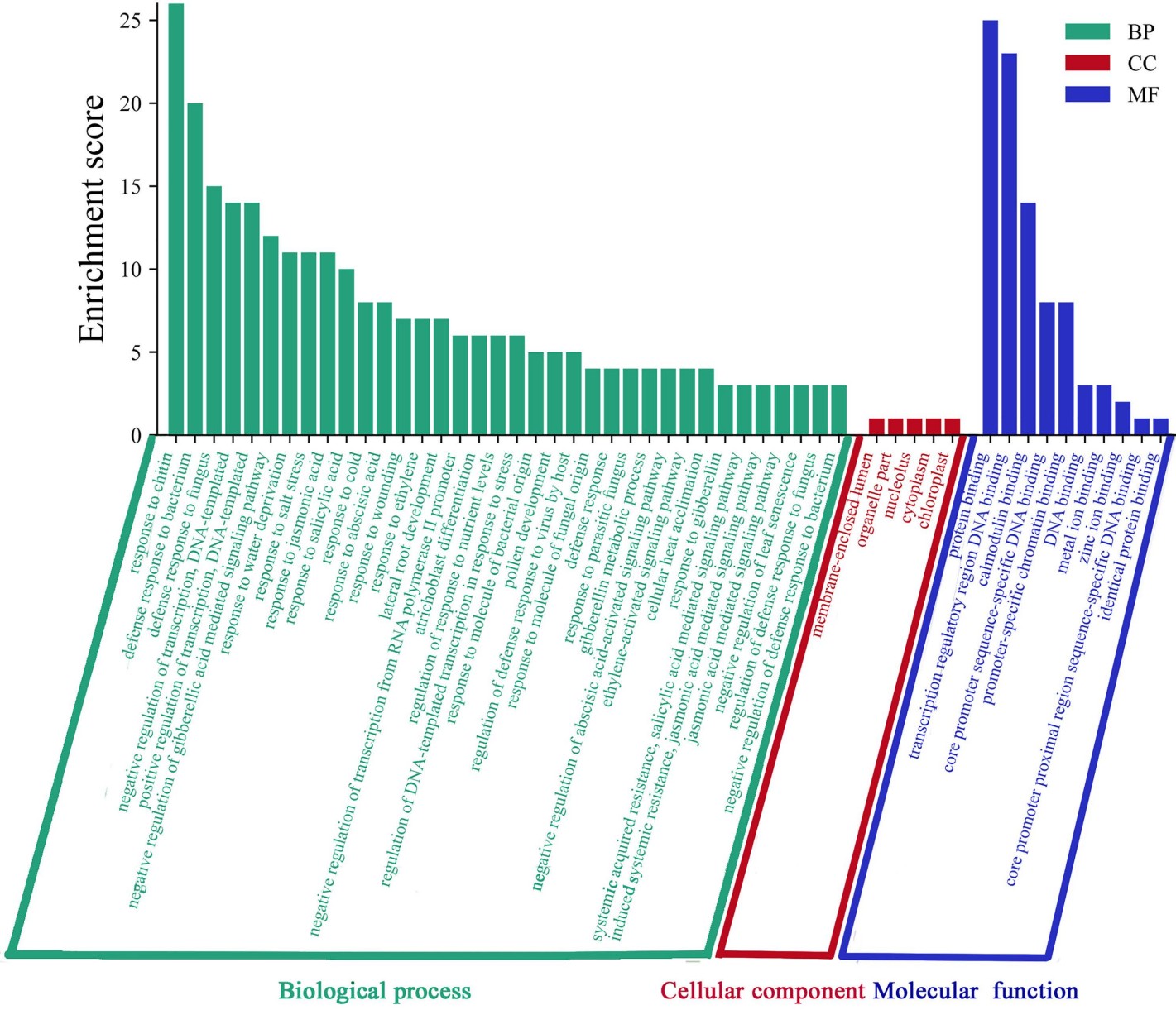

**Fig 6. *PsnWRKYs* gene ontology (GO) enrichment.** The green module represents biological process (BP), the red module represents cellular component (CC), and the blue module represents molecular function (MF).

80 miRNAs formed 214 target relationship pairs. Among them, PtWRKY7, PtWRKY39, PtWRKY64, and PtWRKY95 each correspond to five target miRNAs, while PtWRKY61 and PtWRKY71 correspond to six target miRNAs respectively. These genes may have important function (Fig 7).

### Expression response of *WRKYs* to Cd stress

The spatiotemporal expression patterns revealed that the *WRKYs* responded to Cd stress at different time points and in various tissues. During the Cd treatment, *PtWRKY4*, *PtWRKY65*, and *PtWRKY96* exhibited consistent expression

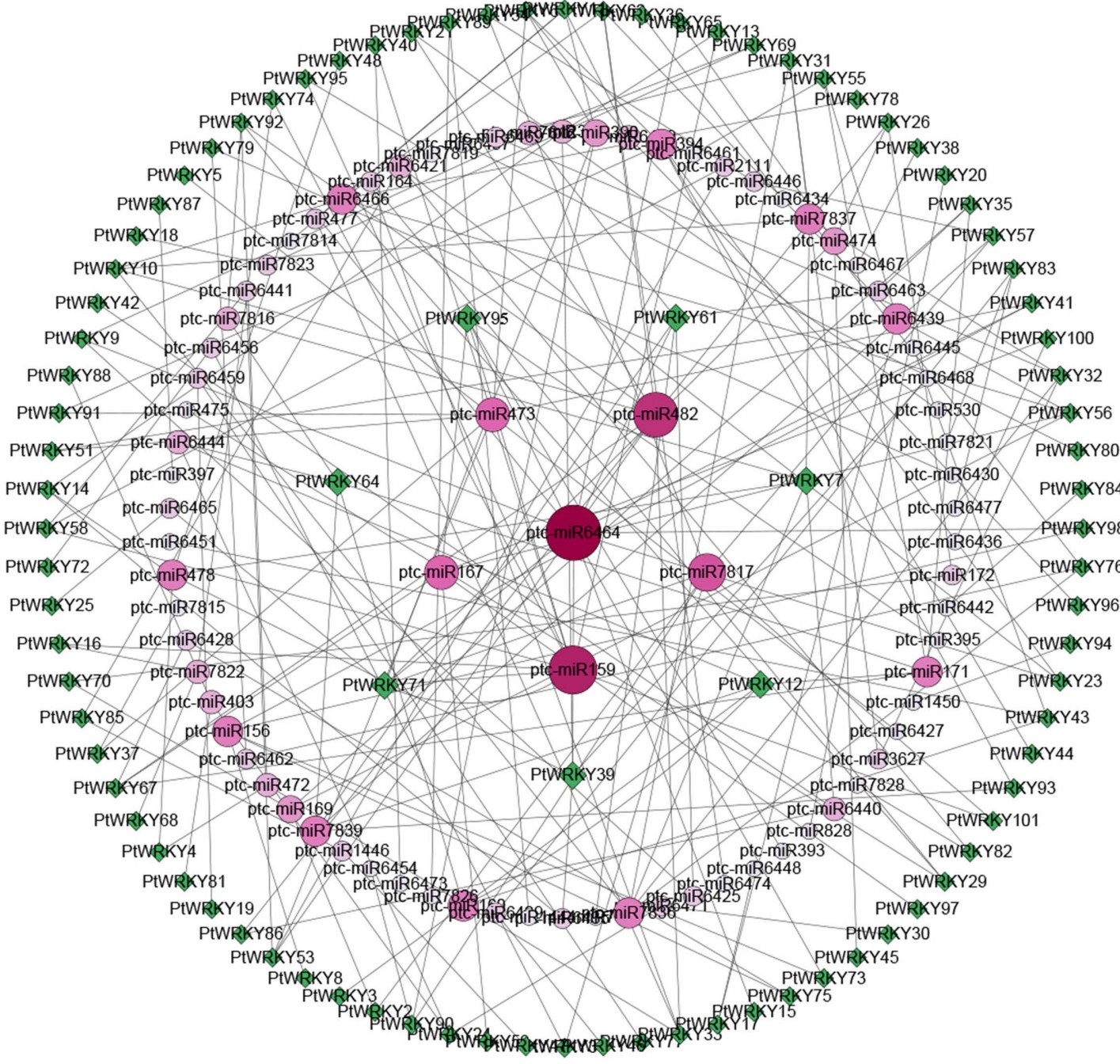

**Fig 7. Analysis of the targeting relationship between PtWRKYs and PtmiRNAs.** The green icon represents PtWRKYs, while the dark pink, pink, light pink, and gray icons all represent miRNAs.

patterns, showing an increasing trend from 0 to 36 hours in roots, while in leaves, their expression continuously increased from 0 to 12 hours, peaking at 12 hours, followed by a decreasing trend from 12 to 24 hours, and then increasing again from 24 to 36 hours. The expression patterns of *PtWRKY19* and *PtWRKY84* were similar, showing an initial increase

followed by a decrease in roots from 0 to 36 hours, while in leaves, their expression increased steadily from 0 to 12 hours, decreased from 12 to 24 hours, and then increased again from 24 to 36 hours. The expression patterns of *PtWRKY38* and *PtWRKY58* were consistent, showing an initial increase followed by a decrease in both roots and leaves within 0–36 hours, reaching peak levels at 12 hours. *PtWRKY62* and *PtWRKY74* exhibited similar patterns, with expression in roots decreasing continuously from 0 to 12 hours, increasing from 12 to 24 hours, and then decreasing again from 24 to 36 hours, while the expression showed the trend of initial decrease followed by an increase in leaves within 0–36 hours (Fig 8).

To investigate the expression patterns of *PtWRKYs* under cadmium stress, we conducted RNA-Seq analysis after applying 150 µM CdCl$_2$ stress to poplar for 24 hours. The results showed that 74 *PtWRKYs* in roots responded to Cd, with 35 up-regulated and 39 down-regulated, showing fold changes of 2.1 to 92.1 for up-regulation and 2.1 to 36.2 for down-regulation. 23 *PtWRKYs* in stems responded to Cd, with 13 up-regulated and 10 down-regulated, showing fold changes of 2.3 and 2.6 for up-regulation and 2.1 to 23.3 for down-regulation. 28 *PtWRKYs* in leaves responded to Cd, with half showing up-regulation and half down-regulation, with fold changes of 2.1 to 482.8 for up-regulation and 2.1 to 37.9 for down-regulation (S8 Table). The Venn diagram indicates that 7 *PtWRKYs* were differentially expressed in all three tissues, with 3 *PtWRKYs* up-regulated and 2 *PtWRKYs* down-regulated across three tissues (Fig 9A). Subsequently, we validated the RNA-Seq results of 12 significantly responsive genes to Cd stress using RT-qPCR, which corroborated the findings. *PtWRKY4*, *PtWRKY19*, *PtWRKY65*, *PtWRKY95*, and *PtWRKY96* were up-regulated by more than 15 folds in the roots, and among them, *PtWRKY65*, *PtWRKY95*, and *PtWRKY96* were up-regulated by over 5 folds in stems and more than 3 folds in leaves (Fig 9B). In summary, we selected *PtWRKY95*, which was significantly upregulated in three tissues, for further study.

## Bioinformatics and subcellular localization analysis of transcription factor PsnWRKY95

We cloned the PsnWRKY95 CDS with a length of 816 bp from *Populus simonii × P. nigra*, encoding 272 amino acids with a highly conserved WRKY domain (Fig 10B), and its protein sequence similarity to PtWRKY95 was 97.05%. It was found through STING analysis that the PsnWRKY95 can interact with eight WRKY family proteins (Potri.013G090300.1, Potri.006G263600.1, Potri.001G002400.1, Potri.018G019800.1, Potri.003G182200.1, Potri.006G109100.1, Potri.005G085200.1, Potri.016G137900.1), as well as with one FAD/NAD(P)-binding oxidoreductase family protein (Potri. T026100.1) and one unknown protein (Potri.003G222500.1). The local clustering coefficient is 0.778, with an enrichment p-value of 1.9e-11 (Fig 10A). Through transient infiltration of tobacco leaves, we analyzed the subcellular localization of the PsnWRKY95 in living cells. PBI121-PsnWRKY95-GFP exhibits green fluorescence exclusively in nucleus, while the control group pBI121-GFP is expressed throughout the entire cell, indicating that PsnWRKY95 is a nuclear-localized protein (S1 Fig).

## Interaction analysis between PsnWRKY95 and Cd resistant cis-acting element

Numerous studies have confirmed that plant transcription factors can specifically bind to the cis-acting element G-box in the promoters of downstream target genes, thereby regulating the downstream target genes and exerting cadmium resistance function. In this study, the significant Cd stress-responsive PsnWRKY95 were selected for yeast one-hybrid experiment. The result indicate that all combinations were able to grow normally on the deficient medium lacking leucine (SD/-Leu), but only the combinations AD-Rec-PsnWRKY95/AB-G-box and the positive control could grow normally on the SD/-Leu medium containing 500 ng/mL Aureobasidin A (AbA) (Fig 11). Therefore, PsnWRKY95 may regulate the Cd resistance function in poplar by specifically binding to G-box in the promoter region of downstream target genes.

## Breeding of the overexpression *PsnWRKY95* tobacco and cadmium resistance function analysis

Using the leaf disk method, we obtained five *PsnWRKY95* transgenic tobacco lines (S2 Fig). RT-qPCR results showed that the *PsnWRKY95* gene was overexpressed in T lines. We then selected T-1, T-4, and T-5, which exhibited higher

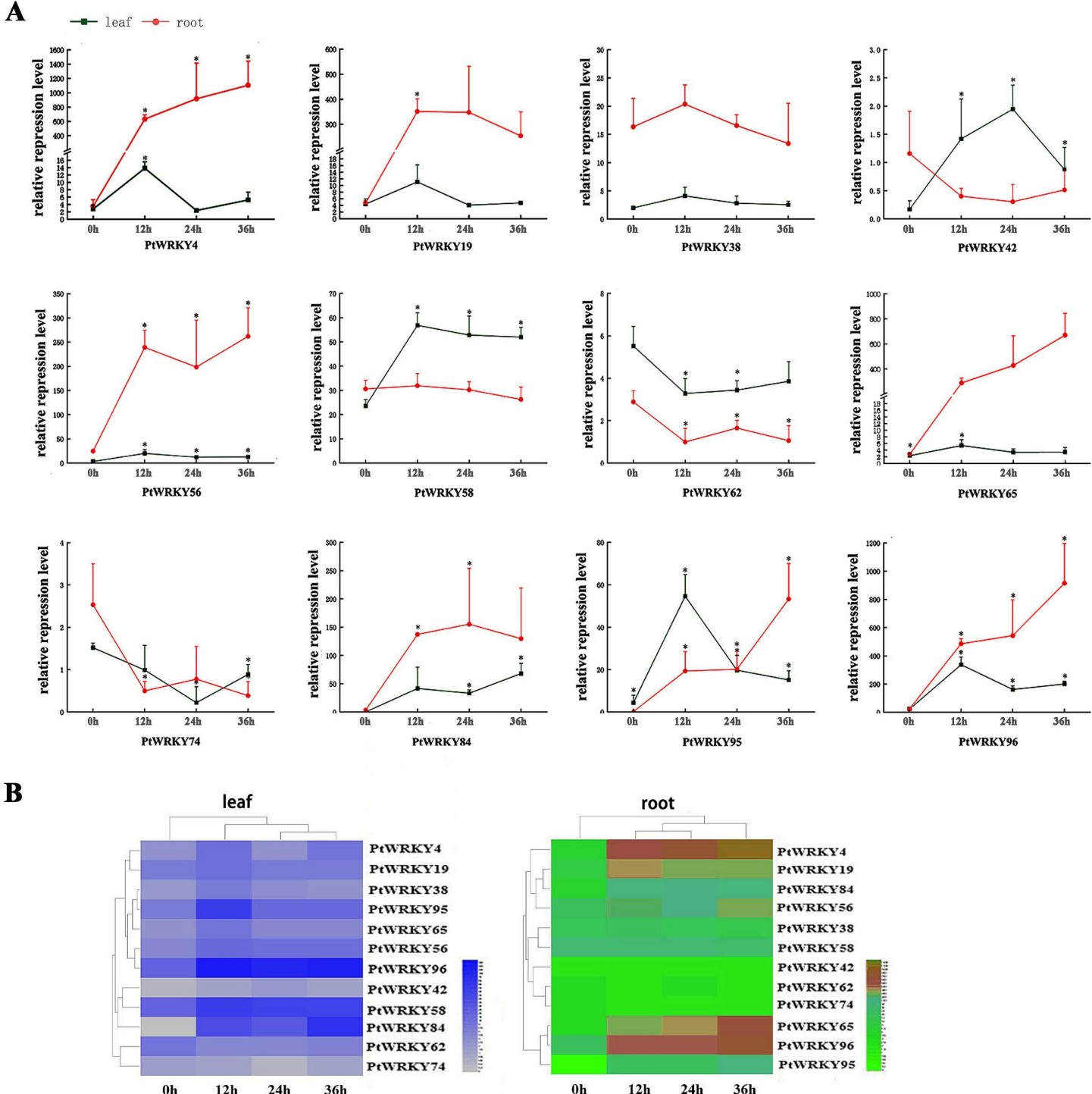

**Fig 8. Analysis of the temporal and spatial expression patterns of *PtWRKYs*.** (A) Line graph of *PtWRKYs* expression after 0h, 12h, 24h, and 36h of Cd stress in poplar, where red and green respectively represent the expression in roots and leaves. (B) Heatmap of *PtWRKYs* expression at different time points.

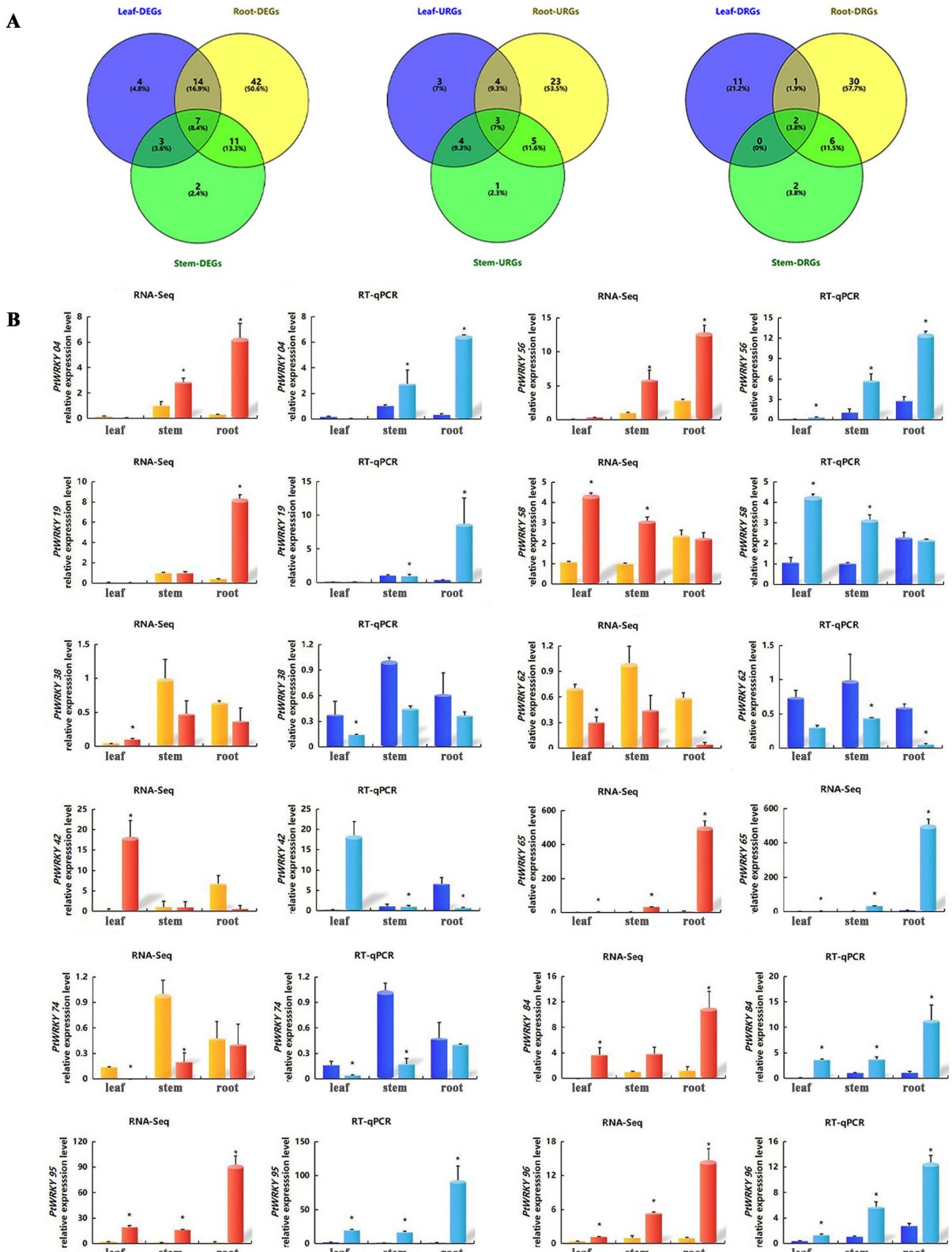

**Fig 9. Analysis of *PtWRKYs* expression in response to Cd.** (A) Venn diagram of DEGs in tissues under Cd stress. The diagram shows the number of DEGs responding to Cd in roots, stems, and leaves, the number of up-regulated DEGs, and the number of down-regulated DEGs. DEG represents

differentially expressed genes, URG represents up-regulated differentially expressed genes, and DRG represents down-regulated differentially expressed genes. (B) Expression levels of *PtWRKYs* in different tissues after Cd stress were analyzed based on RNA-Seq and RT-qPCR. Orange and dark blue represent control treatment, while red and light blue represent Cd treatment. The relative expression levels of each gene under Cd stress were calculated based on the expression of corresponding genes in the control treatment roots. The error bars represent the standard deviation (SD) of biological replicates. "*" indicates significant difference ($p < 0.05$).

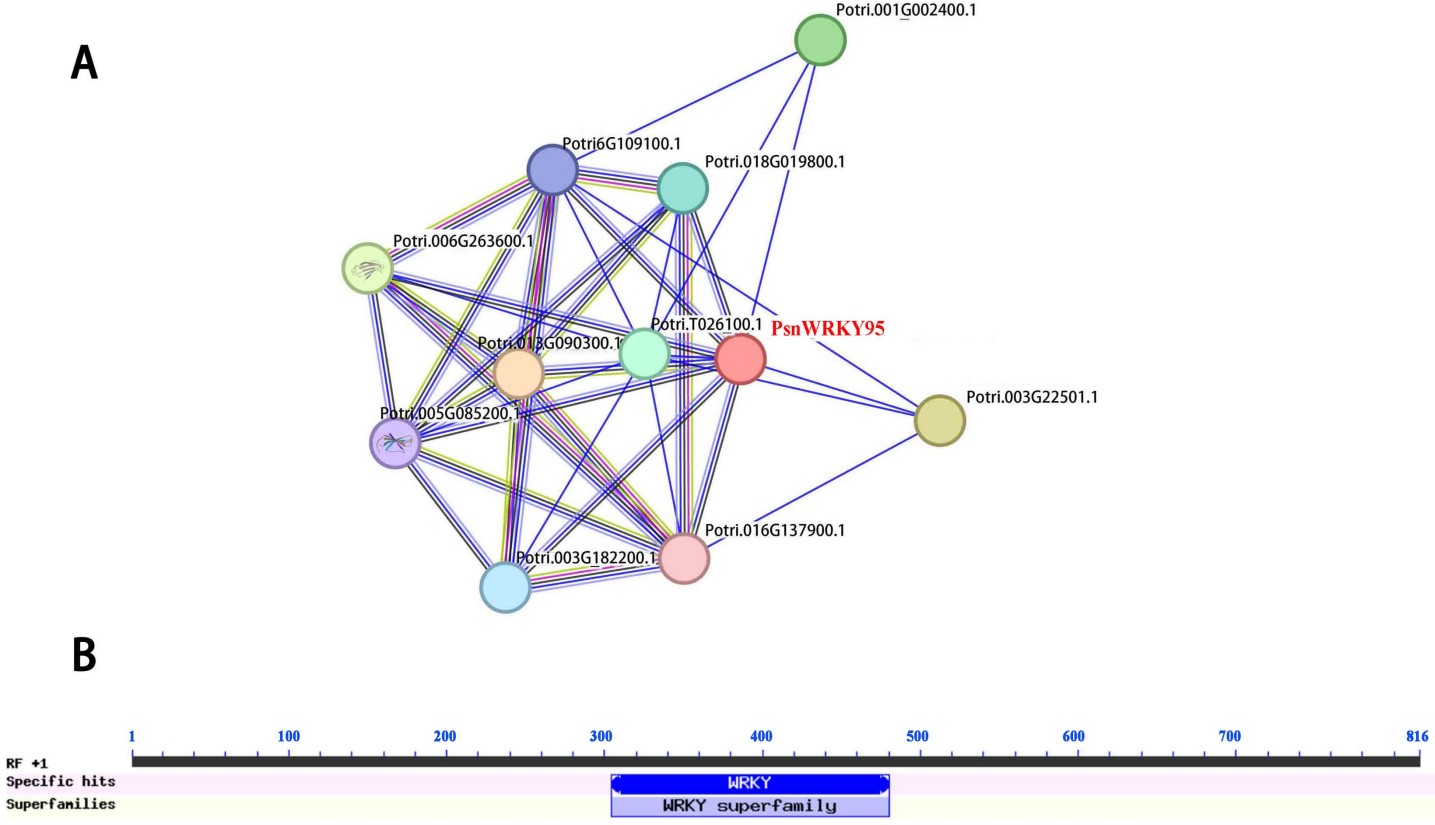

**Fig 10. Bioinformatics and subcellular localization analysis of PsnWRKY95.** (A) Prediction of interaction proteins of PsnWRKY95. The WRKYs interaction protein network is predicted using the online tool STRING, with the target proteins PsnWRKY95 highlighted in red. (B) Display of conserved domains of PsnWRKY95.

expression levels, for functional analysis (Fig 12). Morphologically, there was no significant difference between the WT and transgenic lines under normal condition. However, after 7 days of 150 µM CdCl$_2$ stress, all plants exhibited growth restriction and damage, but the WT leaves showed obvious damaging and wilting, while the transgenic lines displayed less damage (Fig 12D). Additionally, we recorded various growth indices; under normal conditions, there were no significant differences in plant height or root length. After Cd stress, the plant height of the transgenic lines increased by 16%−26% compared to WT, and the root length increased by 12%−27% (Fig 12B). POD activity indirectly reflects the plant's ability to eliminate harmful ROS. Physiological analysis indicated that there were no differences in POD activity among the plants under normal conditions. After Cd stress, the POD activity in transgenic lines were 28%−51% higher than WT (Fig 12C). Moreover, RT-qPCR analysis revealed that POD expression in transgenic lines higher than WT (S3 Fig). This suggests that the *PsnWRKY95* transgenic lines have a better capacity to eliminate reactive oxygen.

Furthermore, under normal condition, MDA content, chlorophyll content and electrolyte leakage rate showed no significant changes among the various plants. After stress, the MDA content in T lines decreased by 13%−32% compared to WT, while chlorophyll content in T lines increased by 15%−29%, and electrolyte leakage rate in T lines decreased by 9%−20% compared to WT (Fig 12C). Additionally, the expression of HMA1, which is associated with Cd resistance, was higher in T lines than WT. These results indicate that the overexpression *PsnWRKY95* plants confer better Cd resistance compared to WT.

## Discussion

WRKY transcription factors are one of the largest gene families in plants, and numerous studies have shown their important roles in plant growth, development, and response to abiotic stress. As yet, the identification of WRKY family members has been carried out in many plants, and the specific molecular function of certain members have been further validated. *Arabidopsis thaliana* has 74 WRKY members, and *AtWRKY71* was found to have an antagonistic effect on delayed flowering in response to salt stress, shortening the life cycle under high salt stress [42]. *Oryza sativa* has 102 WRKY members, *OsWRKY31* forms a MKK10–2/MPK3 functional module with the MAP kinase in immune system, contributing to plant resistance against blast fungus [43]. *Glycine max* has 182 WRKY members, GmWRKY142 can directly bind and activate the Cd tolerance1-like genes (GMCDT1−1 and GMCDT1−2), reducing Cd absorption and regulating Cd tolerance actively [44]. We identified 102 WRKY members in poplar genome. WRKY protein typically contain one or two WRKY domains, which are characterized by a conserved domain WRKYGQK along with a zinc finger motif (C2H2 or C2HC), forming approximately 60 amino acids sequence. PtWRKYs protein contain a C2H2 zinc finger structure at the C-terminus and a highly conserved WRKY domain (WRKYGQK) at the N-terminus, which is consistent with the structural patterns of WRKY protein in *Vitis vinifera*, *Oryza sativa*, and *Salvia miltiorrhiza* [45,46]. All PtWRKYs possess the conserved WRKY domain and are hydrophilic and nuclear localization. Among them, PtWRKY6 and PtWRKY72 also localize to the cytoplasm, PtWRKY90 also localizes to the chloroplast, and PtWRKY5 also localizes to the mitochondria, which indicates that these genes may have transmembrane structure (Fig 1 and S3 Table).

Similar to most plant WRKY families, PtWRKYs can be divided into three groups: Group I, Group II, and Group III, with Group II being the largest, consisting of 55 members, and Group I being the smallest, consisting of 22 members (Fig 2 and S4 Table). The results of this study showed members of the same group showed similar gene structure, CDS length and distribution of motifs, indicating that the evolution and structural variation of PtWRKYs are closely related with the diversity of gene structure. Analysis of the protein motifs and intron patterns can help identify divergent domains and further confirm the results of phylogenetic analysis [47]. The results of this study show that PtWRKY85 in Group I have seven introns, suggesting that it may have undergone complex processes of gene segment splicing or insertion during evolution. All WRKY protein contain motif 1, motif 2, and motif 3, which are likely the main components of the WRKY domain. Members of subgroup IIb, subgroup IIC, subgroup IId, and subgroup IIe also contain additional motif 4, motif 5, and motif 7, as well as special domains such as Plant_zn_clust, TFCD_C superfamily, Uso1_p115_C, Nucleoporin_FG2, BRLZ, and LGIC_ECD. The BRLZ domain of transcription factors can bind to specific DNA sequence to regulate gene expression, indicating that members of Group II may possess new specific function compared to other protein, which require further exploration (Fig 3) [48].

Gene segmental duplication provides the genetic resources for natural selection and is mainly driven by the reproduction of new genes and the expansion of gene families, effectively supporting the adaptation of organisms to different complex environments. Conventional gene duplication includes three modes: segmental duplication, tandem duplication, and transposed event gene duplication [49]. In *Zea mays* WRKY family study, 52 gene duplication involving 78 *ZmWRKYs* were identified, all of which were segmental duplications without tandem duplication [11]. In *Trifolium repens* WRKY family, a total of 124 gene duplication were identified, including 118 segmental duplication and 6 tandem duplication [50]. In this study, 101 *PtWRKYs* formed 160 segmental duplication and one tandem duplication. Segmental duplication dominated

the expansion of the poplar WRKY family (Fig 4A). By calculating the nonsynonymous (Ka) and synonymous (Ks) substitution rates, it was found that all Ka/Ks values were much less than 1, indicating strong purifying selection and removal of deleterious variations in poplar WRKY family, maintaining functionality to a large extent and reducing the occurrence of harmful mutations after duplication (S5 Table).

In the plant kingdom, each gene family originates from the common ancestor and undergoes multiple rounds of evolution to expand. Comparative genomics analysis of collinearity is a rapid and effective method that transfers genomic knowledge obtained from one taxonomic unit to another, inferring the genomic structure and function of the latter. In this study, we compared the homologous genes in *Arabidopsis thaliana* model plants and *Oryza sativa* to predict the function of *PtWRKYs*. The results showed that 72 *PtWRKYs*, 38 *AtWRKYs*, and 18 *OsWRKYs* formed 104 homologous gene pairs, including 72 *PtWRKYs-AtWRKYs* pairs and 32 *PtWRKYs-OsWRKYs* pairs (Fig 4B and S6 Table). It has been reported that *Os01g0289600*, *Os01g0624700*, *Os01g0656400*, and *Os02g0181300* play important roles in the crosstalk between abiotic and biotic stress signaling pathways through mediating ABA. Therefore, their homologous genes *PtWRKY20*, *PtWRKY78*, *PtWRKY17*, *PtWRKY4*, *PtWRKY19*, *PtWRKY65*, *PtWRKY77*, and *PtWRKY95* may have similar function [51]. *AT1G13960.1*, which is co-associated with *PtWRKY58* and *PtWRKY90*, is tolerant to Me-JA stress [52]. *AT1G62300.1* is the homologous gene of *PtWRKY30*, can promote the expression of AKT1 in *Arabidopsis* to regulate potassium acquisition and coordinate the uptake and detoxification of arsenate [53,54]. *AT1G80840.1* is the homologous gene of *PtWRKY2*, plays a crucial role in Fe response pathway [55]. *AT2G44745.1* is the homologous gene of *PtWRKY16*, negatively regulates Fe entry into seeds and inhibits GSH1 expression to negatively regulate Cd tolerance in *Arabidopsis* [23,56]. It is found that the orthologous gene *AtWRKY65* (*AT1G29280.1*) of *PtWRKY31* can respond to the cadmium toxicity, and the orthologous gene *AtWRKY31* (*AT4G22070.1*) of *PtWRKY81* can respond to the cesium toxicity [57,58]. *AtWRKY13* (*AT4G39410.1*) enhances cadmium tolerance by promoting the production of D-cysteine desulfhydrase and hydrogen sulfide [27], *AtWRKY33* (*AT2G38470.1*) plays a crucial role in blocking Cd absorption to prevent metal toxicity in *Arabidopsis* [59], and *AtWRKY45* (*AT3G01970.1*) positively regulates Cd tolerance by activating the expression of PCS1 and PCS2 [24]. This suggests that their respective homologous genes, *PtWRKY37*, *PtWRKY87*, and *PtWRKY66*, may confer Cd tolerance.

The promoter of PtWRKYs contain numbers cis-acting elements that respond to stress and defense reaction, suggesting that these genes may participate in multiple signaling pathways to combat abiotic stress (Fig 5 and S7 Table). In addition to conventional transcriptional regulation activities, most *PtWRKYs* also participate in stress defense molecular function, such as response to abscisic acid (GO:0009737), water deprivation (GO:0009414), and salicylic acid (GO:0009751) (Fig 6). MiRNA are important regulatory factors in plant growth and development, and more than half of miRNA target are transcription factors, such as MYB, SBP, WRKY, HD-ZIP, and AP2 [52,53]. In this study, *PtWRKYs* also bind to numerous miRNAs. ptc-miR1444 and ptc-miR1450 have been reported to respond significantly to abiotic stress, while ptc-miR169 integrates abscisic acid (ABA), jasmonic acid (JA), salicylic acid (SA), and redox signaling in response to many biotic/abiotic stressors in plants, suggesting that their target genes *PtWRKY2*, *PtWRKY35*, *PtWRKY56*, *PtWRKY70*, *PtWRKY89*, and *PtWRKY94* may have similar function [60,61]. It has been reported that ptc-miR2111, ptc-miR6456, and ptc-miR6464 respond positively to heavy metal stress, with ptc-miR2111 playing an important role in the transport of Co, Fe, and Mn, indicating that their target *PtWRKYs* also play a positive role in the transport of heavy metal (Fig 7) [62,63].

In this study, we investigated the response of *PtWRKYs* to Cd stress. A total of 83 *PtWRKYs* responded to Cd stress, and seven *PtWRKYs* showed differential expression in different tissues of poplar, with *PtWRKY84*, *PtWRKY95*, and *PtWRKY96* up-regulated in root, stem, and leave, while *PtWRKY62* and *PtWRKY74* were down-regulated (Fig 9 and S8 Table). The spatial-temporal expression pattern showed that most WRKY genes responded to Cd stress at each time point, but their expression patterns were different, mainly showing peak-valley or fluctuating trends. Among them, *PtWRKY19*, *PtWRKY38*, and *PtWRKY58* showed peak-valley trends in root and leave, while *PtWRKY4*, *PtWRKY65*,

*PtWRKY95* and *PtWRKY96* showed upward fluctuating trends in root (Fig 8). It indicates that these genes may play important roles in poplar resistance to Cd stress and further exploration is necessary.

We cloned *PsnWRKY95* from *Populus simonii × P. nigra*, which belongs to the poplar WRKY Group IIa subfamily and possesses a conserved WRKY domain (Fig 10B). The PsnWRKY95 protein is accurately localized to the nucleus, which aligns with the characteristics of transcription factors (S1 Fig). Salicylic acid (SA) and jasmonic acid (JA) are two important plant hormones that play crucial roles in abiotic stress [64]. The interacting protein of PsnWRKY95, Potri.005G085200.1, functions as a positive regulator of salicylic acid (SA) mediated signal transduction and a negative regulator of jasmonic acid (JA) mediated signal transduction in defense response [65]. The interacting protein of PsnWRKY95, Potri.006G263600.1, mediates abiotic stress through the ABA pathway [66], while the interacting protein Potri.003G182200.1 mediates copper homeostasis through ABA signaling (Fig 10A) [67]. In conclusion, PsnWRKY95 may have similar functions or synergistically form a dimeric structure with the above homologous protein to promote poplar resistance to Cd. A large number of studies have shown that when plants are subjected to metal stress, transcription factors (TFs) can specifically bind to G-box cis-acting element in the promoter to activate target genes so as to resist Cd pressure. GhMYB44 can activate the metal transport protein GhHMA1 by directly binding to the G-box cis-element in the promoter, thereby enhancing *Gossypium* tolerance to $Cd^{2+}$ [68]. Yeast one-hybrid results show that PsnWRKY95 can specifically bind to G-box element, indicating that it may regulate the expression of downstream target genes by binding to their promoters, enhancing the cadmium resistance (Fig 11).

In this study, we obtained five *PsnWRKY95* overexpressing tobacco lines. Generally, there was no morphological difference between the transgenic lines and WT; however, under Cd stress, the growth of the transgenic lines was better than that of WT (Fig 12D). When plants are subjected to external stimuli, they generate reactive oxygen species (ROS) which can be harmful. Peroxidase (POD) is an antioxidant enzyme that protects plant cells from damage and efficiently eliminates ROS [69]. In our study, the POD activity of T-1, T-4, and T-5 was higher than WT after experiencing Cd stress, and the expression level of POD in these lines was also greater than WT, indicating that *PsnWRKY95* overexpressing plants have better ROS clearance ability (Figs 12C and S3). Electrolyte leakage rate and malondialdehyde (MDA) content are indicators for assessing the degree of cellular damage. After seven days of Cd stress, the electrical conductivity and MDA content of *PsnWRKY95* overexpressing lines were lower than WT (Fig 12C), suggesting that the transgenic lines have poorer membrane permeability and less damage. Chlorophyll content can serve as indicators of plant resistance. After Cd stress, the chlorophyll content in *PsnWRKY95* overexpressing lines was higher than WT, indicating stronger resistance in transgenic lines. In the above results, the trends of growth indicators of transgenic plants, as well as physiological indicators such as POD, MDA and chlorophyll, are consistent with those of PyWRKY71 and PyWRKY71 transgenic *Populus yunnanensis*, which fully indicates that genes of the WRKY family can improve the Cd resistance of poplar

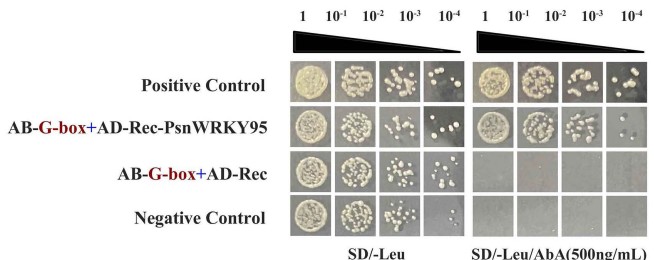

**Fig 11. PsnWRKY95 interact with the Cd resistance cis-acting element.** AB-G-box refers to pAbAi-G-box, AD refers to pGADT7, and AD-PsnWRKY95 refers to pGADT7-PsnWRKY95. SD/-Leu represents the defective medium, and positive transformation was determined after dilution of positive yeast on the Leu plate (500 ng/ml) with Aureobasidin A (AbA). pAbAi-p53/pGADT7-p53 was used as a positive control, and pAbAi-53/pGADT7 was used as a negative control.

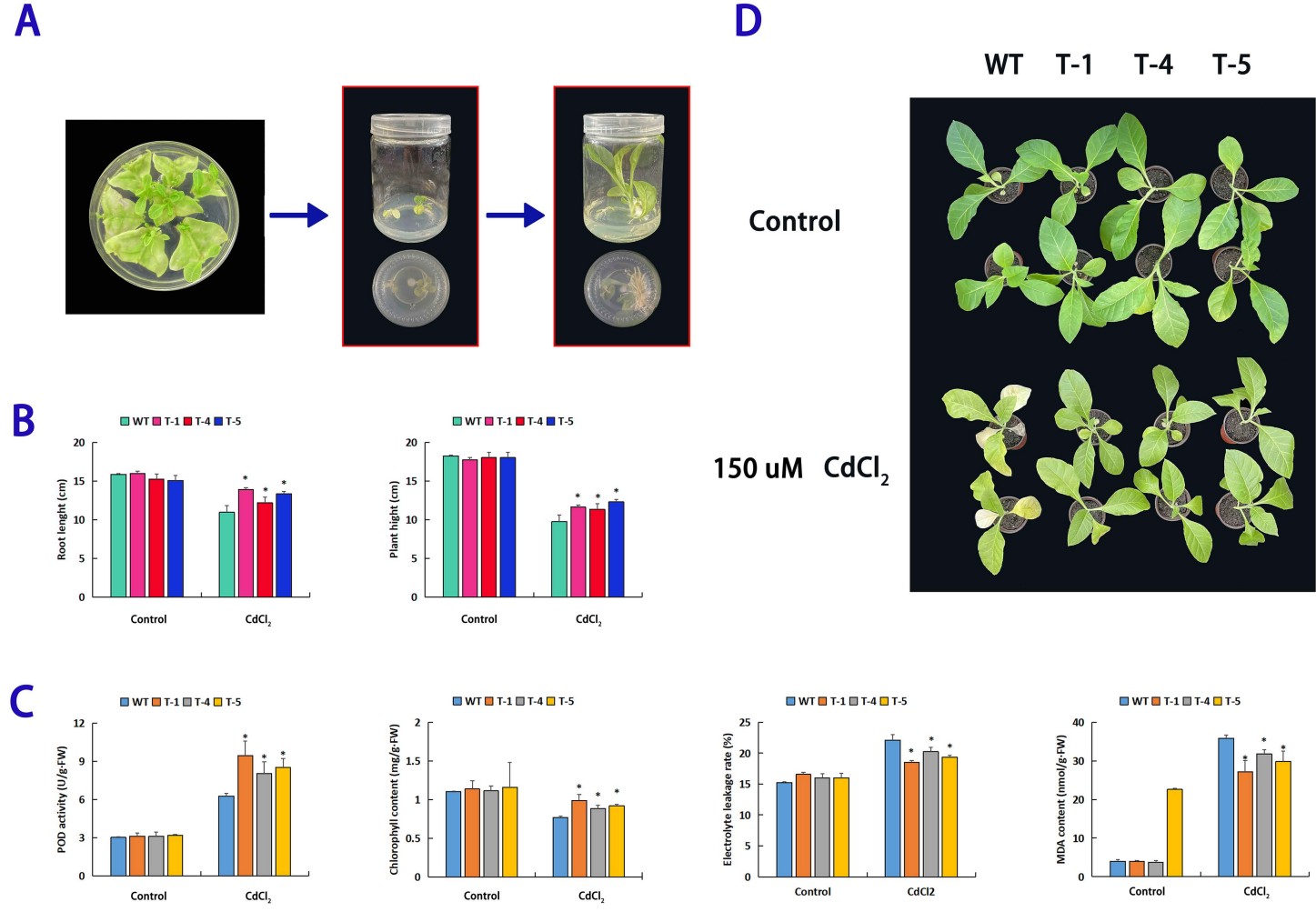

**Fig 12. *PsnWRKY95* transgenic plants selection and functional analysis of *PsnWRKY95* in cadmium resistance.** (A) Screening process of transgenic plants. Rooting culture medium add 50 mg/mL Kan (1 week and 1 month). (B) Growth indicators show root length and plant height from three biological replicates. (C) Physiological indicators showing: peroxidase (POD), malondialdehyde (MDA), chlorophyll content and electrolyte leakage rate, three biological replicates. Wild-type (WT) and *PsnWRKY95* overexpressing tobacco were treated with water and 150μM CdCl₂ for 7 days respectively. The error bars represent the standard deviation (SD) of biological replicates. "*" indicates significant difference ($p < 0.05$).

[70,71].HMA1 is involved in the transport of heavy metals within plants and promotes the accumulation of Cd [72]. In our study, the expression level of HMA1 in the *PsnWRKY95* overexpressing lines was higher than WT (S3 Fig). Based on these findings, it is possible that PsnWRKY95 upregulates HMA1 by specifically binding to the G-box, enhancing the Cd resistance of poplar. Compared with traditional studies, it is of great significance to further explore the cadmium resistance by determining the expression level of the resistance gene HMA1 in transgenic plants and the interaction between Psn-WRKY95 and the stress-resistant element G-box [28]. In summary, *PsnWRKY95* overexpressing lines exhibited better Cd resistance than WT; however, the precise regulatory mechanisms of PsnWRKY95 in response to Cd stress in poplar require further investigation, such as exploring the PsnWRKY95 expression in different tissues of transgenic poplar, and visually observing the accumulation of Cd elements in different poplar tissues using tissue staining technology [70,71]. This will provide a reference for the subsequent introduction of exogenous genes into agricultural crops to enrich heavy metals in non-edible plant tissues.

## Conclusion

This study identified 102 *WRKYs* from poplar genome and conducted a systematic analysis of physicochemical properties, phylogenetics, gene structure, chromosome distribution, evolutionary relationship, promoter cis-acting elements, GO functional annotation and target miRNA. Based on RNA-Seq, this study showed the response of *WRKYs* to cadmium (Cd) stress. Subsequently, in-depth research was conducted on PsnWRKY95, which was significantly up-regulated in root, stem and leave, and its nuclear localization was determined. Using leaf disc method, overexpression *PsnWRKY95* tobacco were obtained. Morphological functions showed that transgenic lines grew better in Cd toxic environment. From a physiological perspective, transgenic lines have more active physiological indices to cope with Cd stress. At the molecular level, the transcription factor PsnWRKY95 specifically binds to the Cd resistance element G-box to regulate the expression of downstream genes. This study aims to further exploring the cadmium resistance mechanism of WRKY genes in poplar, directionally breeding the PsnWRKY95 transgenic poplar with Cd resistance, and utilizing PsnWRKY95 to regulate the downstream HMA1 gene to promote the transport and accumulation of Cd in poplar, thereby reducing soil Cd content and achieving the goal of remediating heavy metal contaminated soil.

## Supporting information

**S1 Table. The primer sequences for RT-qPCR.**
(XLS)

**S2 Table. The primer sequences for yeast one-hybrid experimental vector Construction.**
(XLS)

**S3 Table. Analysis of PtWRKY genes.**
(XLS)

**S4 Table. Phylogenetic tree unit of WRKY proteins in Poplar and *Arabidopsis thaliana*.**
(XLS)

**S5 Table. The Ka/Ks values of PtWRKY paralogous gene pairs.**
(XLS)

**S6 Table. The collinear relationship between PtWRKYs and AtWRKYs/OsWRKYs.**
(XLS)

**S7 Table. Cis-acting elements in the promoter of PtWRKYs.**
(XLS)

**S8 Table. Relative expression level of WRKY genes in different tissues of poplar under cadmium stress.**
(XLS)

**S9 Table. RNA-seq data of WRKY genes.**
(XLS)

**S1 Fig. Subcellular localization of PsnWRKY95.** The pBI121-PsnWRKY95-GFP fusion vector and the pBI121-GFP control vector were transferred into tobacco cells by transient transformation. After transformation for 24–36 hours, the pictures were imaged by Zeiss laser confocal microscope. (1), (4) are GFP fluorescence detection; (2), (5) are bright field; (3), (6) are superposition fields.
(EPS)

**S2 Fig. (A) The PCR detection of the over-expressing *PsnWRKY95* tobacco lines.** (B) The detection of PsnWRKY95 expression level in transgenic tobacco.
(EPS)

**S3 Fig. RT-qPCR analysis showed that the relative expression levels of POD and HMA1 genes in over-expressing *PsnWRKY95* plants.**
(EPS)

## Acknowledgments

We appreciate the experimental platform provided by the State Key Laboratory of Tree Genetics and Breeding at Northeast Forestry University and Heilongjiang University of Science and Technology. We also extend our heartfelt thanks to the editors and reviewers.

## Author contributions

**Conceptualization:** Yuzhao Ma, Qing Guo.

**Data curation:** Yuzhao Ma, Wanying Zhu.

**Formal analysis:** Yuzhao Ma.

**Funding acquisition:** Qing Guo.

**Investigation:** Guoyue Wang, Xiaojin Yang.

**Methodology:** Yuzhao Ma, Yiqi Liu.

**Project administration:** Qing Guo.

**Resources:** Qing Guo.

**Software:** Yuzhao Ma, Fenglin Jia.

**Supervision:** Qing Guo, Hongbo Zhang.

**Visualization:** Yuzhao Ma.

**Writing – original draft:** Yuzhao Ma, Qing Guo.

**Writing – review & editing:** Yuzhao Ma, Qing Guo.

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
