## [Decision Letter · Decision Letter 0]

9 Jun 2025

Dear Dr. Guo,

Thank you for submitting your manuscript to PLOS ONE. After careful consideration, we feel that it has merit but does not fully meet PLOS ONE’s publication criteria as it currently stands. Therefore, we invite you to submit a revised version of the manuscript that addresses the points raised during the review process.

I appreciate the effort put into conducting this comprehensive study. After reviewing the feedback from our reviewers, I would like to highlight several key points that should be addressed to strengthen your manuscript prior to publication.

Please revise the abstract and introduction sections. Including a graphical abstract to visually summarize your findings would be helpful. Additionally, please consider revising the abstract to include more quantitative data and specific details regarding growth parameters and Cd tolerance associated with PsnWRKY95. There is a lack of discussion regarding other published studies on poplar and cadmium stress, which could enhance your comparative analysis and validate the novelty of your research.

The details concerning the selection of plant material, cadmium concentration, and exposure duration require clarification. These elements are vital for replicability and for understanding the experimental design.

Please think about restructuring the results by employing tables or figures to highlight essential expression patterns and connections. Adding real experimental images could further enhance the clarity and transparency of your findings.

There is a lack of clarity between the terms PtWRKY and PsnWRKY, particularly since both types refer to distinct genetic backgrounds. Providing a clear distinction and explanation in the text could reduce confusion.

I recommend carefully addressing all comments and suggestions. Clarifying these points will not only enhance the manuscript's quality but also improve its overall clarity and impact.

Please provide a revised version of your manuscript along with a detailed response to the reviewer’s comments.

We look forward to receiving your revised manuscript.

Kind regards,

Smita Kumar, Ph.D.

Academic Editor

PLOS ONE

Journal Requirements:

Heilongjiang University of Science and Technology Introduction of High-level Talents Scientific Research Start-up Project (HKD202219)

We sincerely thank Heilongjiang University of Science and Technology for providing the start-up fund that made Project HKD202219 possible. We appreciate the experimental platform provided by the State Key Laboratory of Tree Genetics and Breeding at Northeast Forestry University. We also extend our heartfelt thanks to the editors and reviewers.

Heilongjiang University of Science and Technology Introduction of High-level Talents Scientific Research Start-up Project (HKD202219)

Reviewers' comments:

Reviewer's Responses to Questions

**Comments to the Author**

1. Is the manuscript technically sound, and do the data support the conclusions?

Reviewer #1: Partly

Reviewer #2: Yes

Reviewer #3: Partly

2. Has the statistical analysis been performed appropriately and rigorously?

Reviewer #1: N/A

Reviewer #2: Yes

Reviewer #3: I Don't Know

3. Have the authors made all data underlying the findings in their manuscript fully available?

Reviewer #1: Yes

Reviewer #2: Yes

Reviewer #3: No

4. Is the manuscript presented in an intelligible fashion and written in standard English?

Reviewer #1: Yes

Reviewer #2: Yes

Reviewer #3: No

Reviewer #1: After reading this manuscript, which addresses the WRKY transcription factor family in poplar and related cadmium response mechanisms, I provide the following comments:

I suggest incorporating a graphical abstract as well as an abbreviation list;

The abstract could be improved by providing more quantitative information regarding the obtained results and greater specificity in the details. For example, the authors mention improvements in growth parameters and Cd tolerance upon regulation mediated by PsnWRKY95 but do not provide specific details about these aspects;

It would be beneficial to cite more recently published references from 2024 and 2025 throughout the manuscript, including in the introduction;

The introduction could be improved by better separating paragraphs when introducing new ideas—for instance, the introduction of transcription factors (line 53) could begin in a separate paragraph. Additionally, a better contextualization of the biochemical approaches used in the study would help readers understand the methodological expectations, particularly regarding antioxidant molecules;

In line 110, the authors state that "Resistance to heavy metal adversity is crucial for plant survival," but I wonder whether, under real-life conditions, heavy metals actually cause plant death or significant productivity loss in agricultural settings. Personally, I am not aware of major yield losses caused by heavy metals compared to other abiotic factors such as drought, salinity, or extreme temperatures;

Regarding the methodology, details about the selection of plant material, cadmium concentration in the growth medium, and exposure duration to the heavy metal are not clearly provided;

For the results section, I suggest incorporating better organization through tables or figures summarizing the main expression patterns and phylogenetic relationships, which could facilitate data interpretation. Additionally, including real photos and figures from the experiments would enhance transparency for readers;

Importantly, there are other studies closely related to this research topic, also focusing on poplar and cadmium stress. For example:

• "A WRKY transcription factor, PyWRKY75, enhanced cadmium accumulation and tolerance in poplar" - Ecotoxicol Environ Saf. 2022 Jul 1;239:113630. doi: 10.1016/j.ecoenv.2022.113630

• "A WRKY transcription factor, PyWRKY71, increased the activities of antioxidant enzymes and promoted the accumulation of cadmium in poplar" - Plant Physiol Biochem. 2023 Dec;205:108163. doi: 10.1016/j.plaphy.2023.108163

It is unclear why the authors did not consider these studies for a more comparative discussion on WRKY, poplar, and cadmium-related findings. Such a comparative discussion would also help better contextualize the novelty of the current study compared to previous ones;

The discussion could provide a more critical analysis of the study’s limitations, particularly regarding the methodologies used to validate PsnWRKY95’s role in cadmium tolerance and gene expression results. Additionally, it would be relevant to highlight the implications of introducing an exogenous gene under real agricultural conditions;

Conclusion section: The conclusion broadly summarizes the study’s approaches but lacks reflection on the practical implications of the research, as well as a more specific discussion of the obtained quantitative results and insights.

Reviewer #2: The Manuscript “Genome-Wide Identification of the WRKY Gene Family in Poplar and the Positive Role of PsnWRKY95 in Response to Cadmium Stress” is a well written and detailed study of the WRKY transcription factors in the Poplar Genome with a focus in the response to Cd stress. However, some adequations on the text and specially figures are necessary for it to be ready to be published. Many figures are large with very small fonts, making them not very comprehensive for someone reading in a pdf file. Many abbreviations were not defined upon first appearance in the text, some were not given, others only late in the text (eg. SA was defined in line 565 and used in line 93).

The analyses and experiments were well made, but I still had some doubts that could be addressed in the manuscript.

There is some confusion with the names PtWRKY and PsnWRKY: The systematic analysis of physicochemical properties, phylogenetics, gene structure, chromosome distribution, evolutionary relationship, promoter cis-acting elements, GO functional annotation, target miRNA and STRING protein-protein interaction network were all made using the ensembl available poplar genome that is from Populus trichocarpa (Pt) while the expression analysis (RNA-Seq, RT-qPCR), cloning, subcellular localization and overexpression in tobacco were using the hybrid Populus simonii × P. nigra (Psn) genetic background. In some parts of the text they are treated as the same (eg., expression analyses), in others mixed (e.g., STRING analysis). A simple explanation in the text could make the distinction clear, and if there are such low sequence distinctions, in some sections it would be better to just address the genes without a specific suffix, just as WRKY.

By the way, both expression experiments were made with RNA-seq? It was not clear to me for the “temporal” experiment if there were only 12 genes between all 102 that showed the distinction, or that these were previously chosen.

I had this discussion with colleagues before, and you like the authors’ opinion, could not the GFP-tag in the cloned protein C-terminal end affect the function of the studied protein? Could this be relevant to the observed results?

And just another question, just for curiosity, the poplar trees that become more resistant to heavy metal contaminated soil would be bioaccumulators? And how could the accumulated metal affect the destination of these trees? Could the wood, coal or paper be used in the same manner as the “normal” poplar?

More punctual questions and observations can be seen in the pdf file with my comments.

I congratulate the author for their work, and hope to see this manuscript published soon.

Reviewer #3: The work by Ma and coworkers describes the bioinformatics analysis of the WRKY transcription family in poplar, with a wide range of approaches; and continues to determine the expression of these genes with RNA-Seq and RT-qPCR in Cd treated plants. Finally, the authors select and study a Cd-responsive WRKY which they show to provide Cd-resistance to transgenic tobacco plants.

One of my main concern is related to the extensive bioinformatics analysis, which overlaps with previous works, as I mention below, that are not mentioned or contrasted by the authors. There is no clear reason to redo all these analyses and I do not see how these results are novel and better than those published.

Besides this, experimental results are mostly exploratory or weak in relation to the relevance of the selected PtWRKY95 gene, which is only tested in an heterologous system without a clear relevance for the role in using poplar as a Cd accumulator in contaminated settings. Not much study of this gene in poplar is performed beyond gene expression.

Major points

- There are previous works with WRKY TFs in Poplar. He et al. (2012, doi: 10.1007/s00299-012-1241-0) identified 104 WRKY genes. How is the classification in this MS better than that of He et al. Why a new phylogeny is needed and how do they compare. ¿Are gene IDs and names related or linked?

In another work, Chen et al. (2022, doi: 10.3390/genes13122324) presented an analysis of the WRKY family in Poplar, also proposing a phylogeny and studying gene location, collinearity, protein motifs, gene structure and cis-acting elements. How is the new informatio better than this, why is a new analysis relevant or necessary?

- Phylogeny made with NJ (a very simple and limited inference method) and without bootstrapping or any other means of providing support to the nodes

- It is not clearly stated why the authors chose the cross Populus simonii × P. nigra for several experiments and not P. trichocarpa.

- RNA-Seq reads should be deposited in a public repository and an accession to access these data provided.

Minor points

- L65: "modifying plant functional protein to play a miraculous role", please provide a more adequate adjective than miraculous.

- L77: how is that "Many studies have shown that WRKY transcription factors, which can regulate ... are referred to as universal factor." Universal factor?

- P12, L153: Please specify the program used to calculate Ka/Ks substitution rates.

- L176: the DESeq package is mentioned for determining DEGs but no details are given for read processing, mapping and gene count calculation.

- L209: the title "Expression PsnWRKY95 tobacco obtaining and cadmium resistance analysis" is confusing

- L230: please change "clue" to query.

- Figures 8 and 9 have too much white space and is hard to read without zooming in considerably.

- L398: pĺease clarify here, not only in the figure legend, that STINGS was used to say that " PsnWRKY95 (Potri.018G019700.1) can interact with eight WRKY family proteins..."

- Figure 10C: nuclear localization is expected and should be a supplementary figure.

- L416: please correct "Numerous studies have confirmed that the Cd resistant cis-acting element G-box can specifically bind to target genes". The element does not bind target genes.

**Do you want your identity to be public for this peer review?** For information about this choice, including consent withdrawal, please see our Privacy Policy

Reviewer #1: No

Reviewer #2: **Yes: ** Joao B. de Abreu Neto

Reviewer #3: No

---

## [Author Response · Author response to Decision Letter 1]

28 Jul 2025

Dear Reviewers,

Thank you for your comments concerning our manuscript entitled “Genome-Wide Identiffcation of the WRKY Gene Family in Poplar and the Positive Role of PsnWRKY95 in Response to Cadmium Stress" (Manuscript ID: PONE-D-25-02876). Those comments are all valuable and very helpful for revising and improving our paper, as well as the important guiding significance to our researches. We have studied comments carefully and have made correction which we hope meet with approval. Revised portions are marked in yellow in the paper. The main corrections in the paper and the responses to the reviewer's comments are as follows:

Reviewer #1:

1.Response to comment:

I suggest incorporating a graphical abstract as well as an abbreviation list;

Response:

Thank you for your valuable advice. We have included the graphical abstract and abbreviation list in line35-line41 of the manuscript.

2.Response to comment:

The abstract could be improved by providing more quantitative information regarding the obtained results and greater specificity in the details. For example, the authors mention improvements in growth parameters and Cd tolerance upon regulation mediated by PsnWRKY95 but do not provide specific details about these aspects;

Response:

Thank you for your valuable advice. We have included specific details about the growth parameters, cadmium tolerance and quantitative information indicators of PsnWRKY95 transgenic poplars in the abstract.

3.Response to comment:

It would be beneficial to cite more recently published references from 2024 and 2025 throughout the manuscript, including in the introduction;

Response:

Thank you for your careful work. We have incorporated the latest relevant research findings from 2024-2025 into the introduction, such as references 5-8 and 28.

4.Response to comment:

The introduction could be improved by better separating paragraphs when introducing new ideas—for instance, the introduction of transcription factors (line 53) could begin in a separate paragraph.

Response:

Thank you for your valuable advice. We have made the introduction of transcription factors has been placed in a separate paragraph.

5.Response to comment:

Additionally, a better contextualization of the biochemical approaches used in the study would help readers understand the methodological expectations, particularly regarding antioxidant molecules;

Response:

Thank you for your careful work. We have included the specific biochemical methods in Section 2.9, line 248.

6.Response to comment:

In line 110, the authors state that "Resistance to heavy metal adversity is crucial for plant survival," but I wonder whether, under real-life conditions, heavy metals actually cause plant death or significant productivity loss in agricultural settings. Personally, I am not aware of major yield losses caused by heavy metals compared to other abiotic factors such as drought, salinity, or extreme temperatures;

Response:

Thank you for your careful work. Numerous studies show heavy metals cause plant death or reduced agricultural productivity. Heavy metal toxicity (HMT) threatens global agriculture, impairing crop yields via physiological damage, reduced germination, oxidative stress, and inhibited photosynthesis. Cadmium (Cd), accumulating in plants with a 10-30 year half-life, induces root browning, growth inhibition (reduced length/dry weight, impaired lateral roots) and death; it also exacerbates growth inhibition/necrosis by disrupting carbon fixation, chlorophyll levels and photosynthesis, and triggers excessive reactive oxygen species (ROS) via antioxidant system disruption, damaging cells and plants. Soil Cd pollution further reduces produce quality, endangers human health, and undermines agricultural sustainability. We have supplemented this information in line 114 of the introduction.

7.Response to comment:

Regarding the methodology, details about the selection of plant material, cadmium concentration in the growth medium, and exposure duration to the heavy metal are not clearly provided;

Response:

Thank you for your careful work. Regarding the details of this part of the experimental methods, we have provided a detailed explanation in Section 2.5, line 189.

8.Response to comment:

For the results section, I suggest incorporating better organization through tables or figures summarizing the main expression patterns and phylogenetic relationships, which could facilitate data interpretation. Additionally, including real photos and figures from the experiments would enhance transparency for readers;

Response:

Thank you for your valuable advice. Due to the large number of members in the WRKY family genes, the tables or figures illustrating the main expression patterns and phylogenetic relationships are too large to be included in the main text, so they have been placed in the supplementary materials. Additionally, Fig 2, 3, 5, 8, and 9 have been re-adjusted to enhance transparency for readers.

9.Response to comment:

Importantly, there are other studies closely related to this research topic, also focusing on poplar and cadmium stress. For example:

• "A WRKY transcription factor, PyWRKY75, enhanced cadmium accumulation and tolerance in poplar" - Ecotoxicol Environ Saf. 2022 Jul 1;239:113630. doi: 10.1016/j.ecoenv.2022.113630

• "A WRKY transcription factor, PyWRKY71, increased the activities of antioxidant enzymes and promoted the accumulation of cadmium in poplar" - Plant Physiol Biochem. 2023 Dec;205:108163. doi: 10.1016/j.plaphy.2023.108163

It is unclear why the authors did not consider these studies for a more comparative discussion on WRKY, poplar, and cadmium-related findings. Such a comparative discussion would also help better contextualize the novelty of the current study compared to previous ones;

Response:

Thank you for your careful work. This is an excellent and insightful point. These two reports offer a more comprehensive and specific perspective on the research of poplar under cadmium stress, with much to learn from. We have incorporated the findings of these two studies into the discussion section of our research, specifically in line 646.

10.Response to comment:

The discussion could provide a more critical analysis of the study’s limitations, particularly regarding the methodologies used to validate PsnWRKY95’s role in cadmium tolerance and gene expression results. Additionally, it would be relevant to highlight the implications of introducing an exogenous gene under real agricultural conditions;

Response:

Thank you for your valuable advice. By referring to peer research methods on cadmium stress in poplars, we have gained significant insights and recognized the limitations of our research methods. We have analyzed these methodology in the discussion section. Additionally, we have analyzed the benefits of introducing exogenous genes in practical agricultural applications, as detailed in line 653.

11.Response to comment:

Conclusion section: The conclusion broadly summarizes the study’s approaches but lacks reflection on the practical implications of the research, as well as a more specific discussion of the obtained quantitative results and insights.

Response:

Thank you for your careful work. We have revised the conclusion to include practical implications, specific quantitative results, and key insights as suggested, in line 674.

Reviewer #2:

1.Response to comment:

Many figures are large with very small fonts, making them not very comprehensive for someone reading in a pdf file.

Response:

Thank you for your careful work. We have revised the figures, particularly Fig 3, 5, 8, and 9.

2.Response to comment:

Many abbreviations were not defined upon first appearance in the text, some were not given, others only late in the text (eg. SA was defined in line 565 and used in line 93).

Response:

Thank you for your careful work. We have provided comprehensive explanations of abbreviations and included an abbreviation list in manuscript.

3.Response to comment:

There is some confusion with the names PtWRKY and PsnWRKY: The systematic analysis of physicochemical properties, phylogenetics, gene structure, chromosome distribution, evolutionary relationship, promoter cis-acting elements, GO functional annotation, target miRNA and STRING protein-protein interaction network were all made using the ensembl available poplar genome that is from Populus trichocarpa (Pt) while the expression analysis (RNA-Seq, RT-qPCR), cloning, subcellular localization and overexpression in tobacco were using the hybrid Populus simonii × P. nigra (Psn) genetic background. In some parts of the text they are treated as the same (eg., expression analyses), in others mixed (e.g., STRING analysis). A simple explanation in the text could make the distinction clear, and if there are such low sequence distinctions, in some sections it would be better to just address the genes without a specific suffix, just as WRKY.

Response:

Thank you for your careful work. The main experimental subject of this study is Populus simonii × P. nigra. Compared with other poplar varieties, Populus simonii × P. nigra has the characteristics of fast growth, strong adaptability and good cold resistance, and can grow in high-cold regions such as Heilongjiang, China. Through mass selection of improved varieties, regional trials and years of practice, it has been proved to be an excellent fast-growing and high-yielding tree species. However, the genome of Populus simonii × P. nigra is still incomplete and has not been published. Populus trichocarpa, as a model plant among woody plants, is a typical representative tree species for studying forest genetics. Its genome was sequenced and published in 2006, and has been repeatedly corrected, which is relatively mature and effective. We have carried out a large amount of transcriptome sequencing work on Populus simonii × P. nigra. Through comparison, it was found that the gene sequences of Populus simonii × P. nigra and Populus trichocarpa are extremely similar with high homology, both reaching more than 99%. Therefore, in this study, the Populus trichocarpa genome was used as the reference genome to search for genes, and systematic analyses of physicochemical properties, phylogenetics, gene structure, chromosome distribution, evolutionary relationship, promoter cis-acting elements, GO functional annotation and target miRNA were performed. The expression analysis, cloning, subcellular localization, STRING protein-protein interaction network and overexpression in tobacco were conducted using the hybrid Populus simonii × P. nigra genetic background. Among them, the STRING protein-protein interaction network analysis was performed using the PsnWRKY95 sequence obtained by cloning and sequencing. We have corrected the text in Section 2.6 and Figure 10. In addition, the information about the two poplar species mentioned above has been explained in the introduction and Section 2.1 of the methods.

4.Response to comment:

By the way, both expression experiments were made with RNA-seq? It was not clear to me for the “temporal” experiment if there were only 12 genes between all 102 that showed the distinction, or that these were previously chosen.

Response:

Thank you for your careful work. Both expression experiments were conducted RNA-seq. From the results of the spatiotemporal expression pattern experiment, it was found that the majority of WRKY genes respond to cadmium stress. However, presenting the results of all members would make the manuscript excessively lengthy. Therefore, only 12 genes with higher response levels were selected for result presentation. The selection was based on the results of the significant differential expression experiment, 12 genes, including WRKY4, WRKY19, and WRKY38 showed the most significant results.

5.Response to comment:

I had this discussion with colleagues before, and you like the authors’ opinion, could not the GFP-tag in the cloned protein C-terminal end affect the function of the studied protein? Could this be relevant to the observed results?

Response:

Thank you for your careful work. We also had concerns about this issue when we initially conducted the transgenic research. However, based on the research results from myself and colleagues in our group, who have performed genetic transformation and expression experiments of other genes in poplar, the results consistently showed that the GFP tag at the C-terminus of the cloned protein hardly affects the function and phenotype of the target protein. The relevant research findings are as follows:

Guo Q, Wei R, Xu M, Yao W, Jiang J, Ma X, Qu G, Jiang T. Genome-wide analysis of HSF family and overexpression of PsnHSF21 confers salt tolerance in Populus simonii × P. nigra. Front Plant Sci. 2023 Apr 26;14:1160102.

Cheng Z, Zhu Y, He X, Fan G, Jiang J, Jiang T, Zhang X. Transcription factor PagERF110 inhibits leaf development by direct regulating PagHB16 in poplar. Plant Sci. 2025 Jan;350:112309.

Cheng Z, Fan G, Jiang J,et al.Transcription factor PagERF110 inhibits xylem differentiation by direct regulating PagXND1d in poplar[J].Industrial Crops and Products, 2024, 215(000):14.

6.Response to comment:

And just another question, just for curiosity, the poplar trees that become more resistant to heavy metal contaminated soil would be bioaccumulators? And how could the accumulated metal affect the destination of these trees? Could the wood, coal or paper be used in the same manner as the “normal” poplar?

Response:

Thank you for your careful work. Yes, poplar can act as bioaccumulators of heavy metals in soil. Numerous studies have shown that the regulation of related genes in plant transgenic technology can lead to the accumulation of heavy metals in different tissues of trees. For example, PyWRKY75 significantly enhances the cadmium uptake and accumulation capacity of poplars, with the cadmium uptake of the OE-41 line being 51.32% higher than that of the wild type (Xiaolu W, Qi C, Lulu C, et al. A WRKY transcription factor, PyWRKY75, enhanced cadmium accumulation and tolerance in poplar. [J]. Ecotoxicology and Environmental Safety, 2022, 239: 113630). PyWRKY71 significantly promotes cadmium accumulation in poplars, especially in roots the cadmium content in the OE-45 and OE-87 lines reaches 1.42 times that of the wild type (Xiaoxi C, Xiaolu W, Chengyu H, et al. A WRKY transcription factor, PyWRKY71, increased the activities of antioxidant enzymes and promoted the accumulation of cadmium in poplar. [J]. Plant Physiology and Biochemistry : PPB, 2023, 205: 108163). In rice, OsHMA3 sequesters cadmium into the vacuoles of root cells, restricting cadmium transport from roots to shoots and reducing cadmium accumulation in grains; in mutants, cadmium accumulation in roots decreases, while cadmium content in grains increases (Lu C, Zhang L, Tang Z, Huang XY, Ma JF, Zhao FJ. Producing cadmium-free Indica rice by overexpressing OsHMA3. Environ Int. 2019 May; 126: 619-626; Yan H, Jiao X, Chen Y, et al. Knockout of OsHMA3 in an indica rice increases cadmium sensitivity and inhibits plant growth [J]. Plant Growth Regulation, 2024, 103(3): 635-646).

As for whether the wood, coal, or paper derived from these poplars can be used in the same manner as those from "normal" poplars, we are currently conducting poplar transformation of PsnWRKY95, and the these issues will be further studied in subsequent work.

7.Response to comment:

More punctual questions and observations can be seen in the pdf file with my comments.

Response:

Thank you for your careful work. We have revised the issues marked in the manuscript PDF.

Reviewer #3:

1.Response to comment:

One of my main concern is related to the extensive bioinformatics analysis, which overlaps with previous works, as I mention below, that are not mentioned or contrasted by the authors. There is no clear reason to redo all these analyses and I do not see how these results are novel and better than those published.

There are previous works with WRKY TFs in Poplar. He et al. (2012, doi: 10.1007/s00299-012-1241-0) identified 104 WRKY genes. How is the classification in this MS better than that of He et al. Why a new phylogeny is needed and how do they compare. ¿Are gene IDs and

---

## [Decision Letter · Decision Letter 1]

26 Aug 2025

Genome-Wide Identiffcation of the WRKY Gene Family in Poplar and the Positive Role of PsnWRKY95  in Response to Cadmium Stress

PONE-D-25-02876R1

Dear Dr. Guo,

We’re pleased to inform you that your manuscript has been judged scientifically suitable for publication and will be formally accepted for publication once it meets all outstanding technical requirements.

Kind regards,

Smita Kumar, Ph.D.

Academic Editor

PLOS ONE

Additional Editor Comments (optional):

Reviewers' comments:

Reviewer's Responses to Questions

**Comments to the Author**

Reviewer #1: All comments have been addressed

2. Is the manuscript technically sound, and do the data support the conclusions?

Reviewer #1: Yes

3. Has the statistical analysis been performed appropriately and rigorously?

Reviewer #1: N/A

4. Have the authors made all data underlying the findings in their manuscript fully available?

Reviewer #1: Yes

5. Is the manuscript presented in an intelligible fashion and written in standard English?

Reviewer #1: Yes

Reviewer #1: Although I believe that more functional in planta analyses should be conducted, the manuscript has been improved and I have no further comments at this time.

**Do you want your identity to be public for this peer review?** For information about this choice, including consent withdrawal, please see our Privacy Policy

Reviewer #1: No

---

## [Editor Report · Acceptance letter]

PONE-D-25-02876R1

PLOS ONE

Dear Dr. Guo,

I'm pleased to inform you that your manuscript has been deemed suitable for publication in PLOS ONE. Congratulations! Your manuscript is now being handed over to our production team.

Kind regards,

on behalf of

Dr. Smita Kumar

Academic Editor

PLOS ONE